# Molecular basis of Tousled-Like Kinase 2 activation

Gulnahar B. Mortuza[1], Dario Hermida[1], Anna-Kathrine Pedersen[2], Sandra Segura-Bayona [3], Blanca López-Méndez[4], Pilar Redondo [5], Patrick Rüther [2], Irina Pozdnyakova[4], Ana M. Garrote[5], Inés G. Muñoz[5], Marina Villamor-Payà[3], Cristina Jauset[3], Jesper V. Olsen [2], Travis H. Stracker[3] & Guillermo Montoya [1]

Tousled-like kinases (TLKs) are required for genome stability and normal development in numerous organisms and have been implicated in breast cancer and intellectual disability. In humans, the similar TLK1 and TLK2 interact with each other and TLK activity enhances ASF1 histone binding and is inhibited by the DNA damage response, although the molecular mechanisms of TLK regulation remain unclear. Here we describe the crystal structure of the TLK2 kinase domain. We show that the coiled-coil domains mediate dimerization and are essential for activation through ordered autophosphorylation that promotes higher order oligomers that locally increase TLK2 activity. We show that TLK2 mutations involved in intellectual disability impair kinase activity, and the docking of several small-molecule inhibitors of TLK activity suggest that the crystal structure will be useful for guiding the rationale design of new inhibition strategies. Together our results provide insights into the structure and molecular regulation of the TLKs.

[1] Structural Molecular Biology Group, Novo Nordisk Foundation Centre for Protein Research, Faculty of Health and Medical Sciences, University of Copenhagen, Blegdamsvej 3B, 2200 Copenhagen, Denmark. [2] Mass Spectrometry for Quantitative Proteomics, Novo Nordisk Foundation Centre for Protein Research, Faculty of Health and Medical Sciences, University of Copenhagen, Blegdamsvej 3B, 2200 Copenhagen, Denmark. [3] Institute for Research in Biomedicine (IRB Barcelona), The Barcelona Institute of Science and Technology (BIST), C/ Baldiri Reixac, 10, 08028 Barcelona, Spain. [4] Protein Production and Characterization Platform, Novo Nordisk Foundation Centre for Protein Research, Faculty of Health and Medical Sciences, University of Copenhagen, Blegdamsvej 3B, 2200 Copenhagen, Denmark. [5] Macromolecular Crystallography Group, Structural Biology and Biocomputing Program, Spanish National Cancer Research Center (CNIO), c/Melchor Fernández Almagro 3, 28029 Madrid, Spain. These authors contributed equally: Gulnahar B. Mortuza, Dario Hermida. Correspondence and requests for materials should be addressed to G.M. (email: guillermo.montoya@cpr.ku.dk)

The plasticity of chromatin structure is critical for the regulation of essential cellular processes required for normal development and aging, including DNA replication, DNA repair, cell division and transcription[1,2]. The major subunit of chromatin is the nucleosome, a stretch of DNA wrapped around a histone core octamer[3]. Histones are regulated at the transcriptional level and through myriad post-translational modifications (PTMs) that occur both in complexes with histone chaperones and in the context of the nucleosome[4]. Histone chaperones, including ASF1, CAF-1, and HIRA, are required for the dynamic maintenance of chromatin structure and play diverse roles in the regulation of histone PTMs, subcellular localization, storage and deposition[4].

Anti-silencing factors 1a and 1b (ASF1a and ASF1b) are histone H3–H4 chaperones that mediate replication-dependent and -independent pathways of histone exchange[5,6]. In conjunction with other factors, ASF1 enforces transcriptional silencing, facilitates transcription through heterochromatic regions and supports promoter maintenance[7–9]. ASF1 regulates histone usage during DNA replication by promoting the incorporation of parental and newly synthesized histones, as well as buffering histone pools upon replication stress or arrest[4,10,11]. The loss of ASF1 leads to global transcriptional deregulation, DNA replication defects, and results in an activated DNA damage response (DDR), gross chromosomal rearrangements and DNA damage sensitivity to a wide variety of lesions[12–16].

The precise mechanisms by which ASF1 is regulated remain unclear, but ASF1 was identified as a target of the Tousled kinase (TSL) and Tousled-like kinases (TLKs) that can influence its stability and histone binding capacity[17–19]. TLKs belong to a distinct branch of Ser/Thr kinases, that appear to be constitutively expressed in most cell lines and tissues, and they exhibit the highest levels of activity during S-phase[19–21]. An essential role in proper flower and leaf development was described for TSL[22], whereas its homologues in human, mouse, *Drosophila melanogaster*, *Caenorhabditis elegans* and *Trypanosoma brucei*, have been implicated in DNA repair, DNA replication, transcription and mitosis[18,20,23–26]. In mammals, there are two distinct *TLK* genes, *TLK1* and *TLK2*. TLK activity is rapidly attenuated by the activation of the DDR, and in the case of TLK1, this is due to the direct phosphorylation of a C-terminal residue by Checkpoint kinase 1 (CHK1)[25,27]. In addition to ASF1, TLK1 has been reported to phosphorylate RAD9, a component of the RAD9–RAD1–HUS1 (9-1-1) complex that regulates CHK1 activation[28–31], indicating complex coordination between TLK activity and the checkpoint response. Recent work has linked TLK2 activity to DNA damage checkpoint recovery[32] and shown that TLK1 and TLK2 play largely redundant roles in genome maintenance[26]. This is consistent with the fact that they form heterocomplexes, as suggested by previous work[19,33].

At the organismal level, the single *Tlk* gene in *C. elegans* and *Drosophila* is essential for viability[23,24]. *Tlk1*-deficient mice are also viable, while mice lacking *Tlk2* perish during embryogenesis due to placental failure[26]. Bypass of the placenta using a conditional allele allowed the generation of adult *Tlk2* null animals that, like *Tlk1*-deficient mice, showed no overt phenotypes. Copy number amplifications and increased expression of *TLK2* have been recently reported in ER-positive breast cancer[34–37] and *TLK2* mutations have been implicated in intellectual disability (ID) patients[38] suggesting that TLK activity influences human disease.

Here, we report a detailed molecular characterization of the crystal structure of the kinase domain of TLK2 in complex with ATPγS, thus providing insight into its structural properties and mode of activation. Our biochemical analysis has identified key autophosphorylation sites critical for its activity and indicates that TLK2 is activated through a *cis*-autophosphorylation mechanism. Moreover, we examined the effect of *TLK2* mutations reported in ID patients in a structural context and determined how they impair TLK2 activity.

## Results

**Architecture and activity of human TLK2.** Human TLK1 and TLK2 polypeptides are composed of an N-terminal region, a middle region of helices predicted to contain three coiled coils (CC) and a C-terminal kinase domain that shows 94% identity between TLK1 and TLK2 (Fig. 1a, Supplementary Figure 1). The C-terminal regions of the kinase domains contain many possible phosphorylation sites in their carboxy terminus (C-tail), one of which, S743, has been reported to negatively regulate activity in TLK1[18,25,27]. The first 200 amino acids of the N-terminus are predominantly disordered and are predicted to contain a nuclear localization signal (NLS). The CC domains of *Arabidopsis* TSL, which contain an additional insertion in the first CC, have been reported to mediate oligomerization and activity[33]. Sequence alignment from plants to mammals showed that both the kinase and the predicted CC domains are highly conserved (Supplementary Figure 1) and the last residues of the C-tail were predicted to be unstructured. Based on these sequence alignments and secondary structure predictions, we generated a series of constructs with N- and C-terminal deletions to facilitate the expression and analysis of the different sections of the TLK2 protein (Fig. 1a, Supplementary Figure 2). These human TLK2 variants were overexpressed in *Escherichia coli* in the presence or absence of lambda phosphatase in order to generate phosphorylated and unphosphorylated proteins (Supplementary Figure 2 and Methods). Similarly, we also generated versions of the kinase domain with and without the C-tail; KdomL and KdomS. Finally, a catalytically inactive version (kinase dead (KD)) of the constructs was produced by generating the D613A mutation in the catalytic loop (Supplementary Figure 2).

The oligomerization of TLKs has been suggested based on indirect experiments but has not been examined biochemically to date[33]. To determine the oligomerization status of the purified proteins, we performed size exclusion chromatography coupled to a Multiple Angle Laser Light Scattering (MALLS) instrument. The kinase domain constructs behaved as monomers in solution (≈37 kDa) and an increase in hydrodynamic radius was observed in the samples that were not co-expressed with the lambda phosphatase, demonstrating that phosphorylation (indicated as –p) results in a conformational change (Fig. 1b). In contrast, the longer ΔN-TLK2 and ΔN-TLK2s proteins showed a molecular weight consistent with the dimeric form of the kinase (≈137 kDa), providing clear biochemical evidence that TLKs can form dimers and that this is dependent on the region containing the CC domains (Fig. 1c). We have not detected dissociation and interconversion of monomers in the isolated ΔN-TLK2 or ΔN-TLK2s over a time span of several days. While ΔN-TLK2 and ΔN-TLK2s formed perfect dimers when co-expressed with lambda phosphatase, the phosphorylated forms exhibited larger hydrodynamic radii indicating the presence of additional oligomeric species, especially in the case of ΔN-TLK2 containing the C-tail (Fig. 1c).

As previous work had linked the dimerization ability of TSL to its catalytic activity[33], we expressed a series of Strep-FLAG (SF)-tagged TLK2 alleles with mutations in the N-terminus, CC domains or kinase domain in human cells and examined protein–protein interactions and kinase activity in pull-downs with streptavidin (Strep). In Strep pull-downs of tagged wild-type (WT) TLK2, we readily observed co-precipitation of TLK1, ASF1, a key TLK substrate, and LC8, another prominent interactor that

does not appear to be a target of TLK2 activity (Fig. 1d)[26]. All of these interactions were maintained in the N-terminal deletion (ΔN2-TLK2) mutant lacking the first 161 residues, as well as the KD protein. To address the potential roles of the CC domains, we

generated deletion mutants lacking each of the three domains. Deletion of CC1 strongly impaired the interaction with both TLK1 and LC8 but did not appear to influence that of ASF1. In contrast, neither the CC2 nor CC3 domain had a clear influence

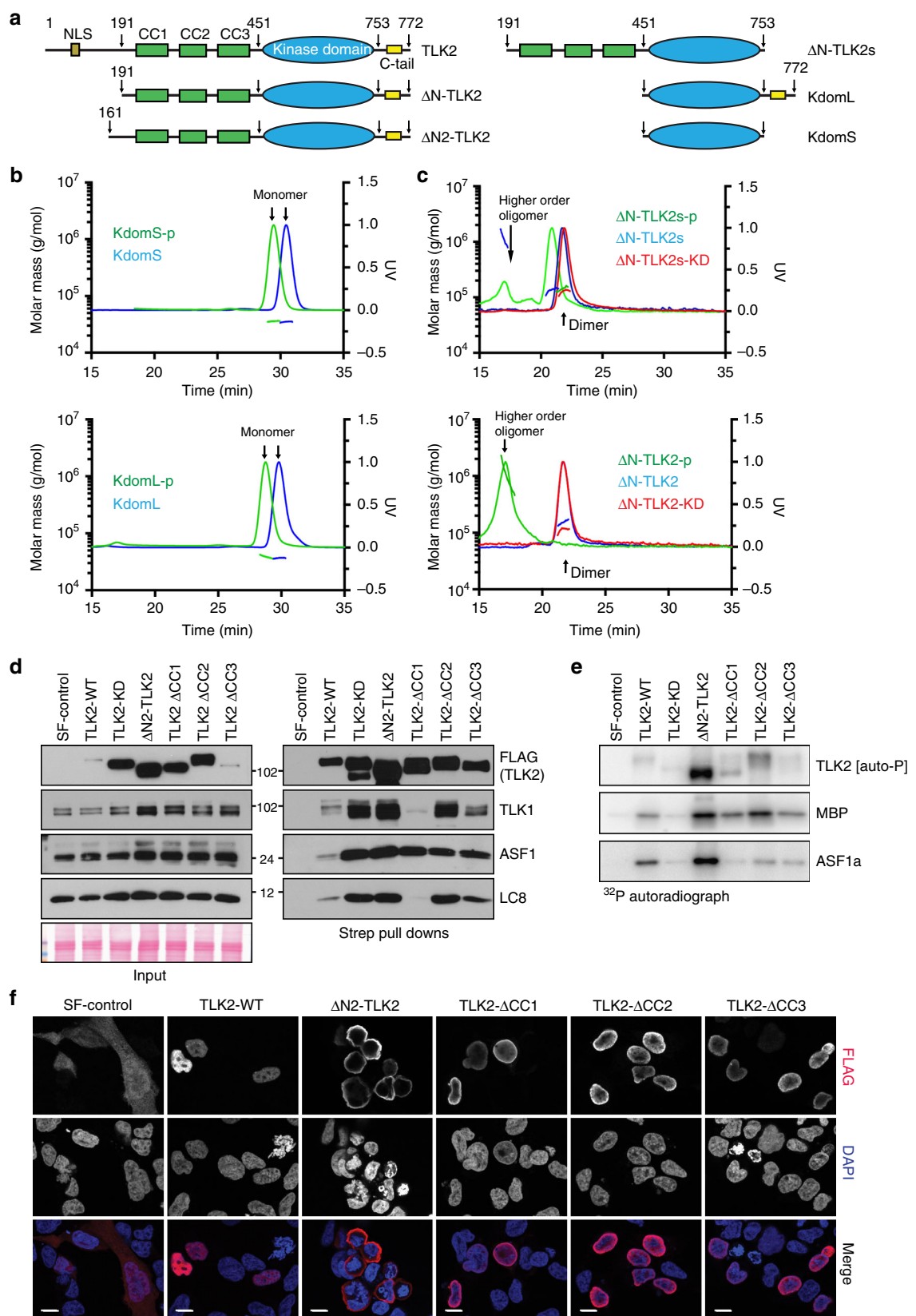

on any of the protein–protein interactions (Fig. 1d). We next examined kinase activity in the pull-downs using Myelin basic protein (MBP), a commonly used kinase substrate surrogate. The effect of the TLK2-KD mutation was clearly evident, as autophosphorylation and substrate phosphorylation were reduced to background levels compared to TLK2-WT (Fig. 1e). The deletion of the N-terminus resulted in highly elevated autophosphorylation and substrate phosphorylation, suggesting that it may play a negative regulatory role and that it is not required for activity. Deletion of any of the individual CC domains did not abolish TLK2 activity, indicating that none of these domains, or the heterodimerization with TLK1, are required for TLK2 activity. We next examined the subcellular localization of these mutants and found that with the exception of the ΔN-TLK2 mutant that contains the putative NLS, all of the mutants showed nuclear localization (Fig. 1f). Therefore, we concluded that TLK2 is a predominantly nuclear protein whose dimerization requires the predicted CC1 domain, and the phosphorylation of the enzyme promotes the assembly of higher order oligomers, likely involving both the CC domains and the C-tail.

**TLK2 auto- and trans-phosphorylation**. Our method to generate the recombinant TLK2 proteins permits the isolation of samples in the presence or absence of autophosphorylations, allowing the study of how these PTMs affect the catalytic properties of the enzyme. A kinase assay was used to measure auto and substrate phosphorylation of the different purified proteins. We first examined the autophosphorylation rates of our various TLK2 constructs over a 7-min time course that was triggered by the addition of ATP (Fig. 2a, Supplementary Figure 3). The unphosphorylated ΔN-TLK2 and ΔN-TLK2s proteins showed a continuous increase in the levels of autophosphorylation over the time course of the experiment, in contrast to their respective phosphorylated forms ΔN-TLK2-p and ΔN-TLK2s-p. Notably, the presence of the C-tail dramatically enhanced autophosphorylation in the presence of the CC domains in the ΔN-TLK2 protein, probably by providing a region rich in phosphorylatable residues that could be involved in kinase regulation. The levels of autophosphorylation of the kinase domain species, KdomS and KdomL, were negligible in comparison to the ΔN-term forms, suggesting that dimerization enhances the activity of the enzyme (Fig. 2b). Moreover, these results show that TLK2 can be activated without the need of an external triggering kinase, as is the case for CDKs, as well as many other kinases[39]. To address whether the autophosphorylation reaction is uni- or bimolecular, we used a competition assay, incorporating different ratios of a ΔN-TLK2-KD mutant (D613A). These experiments revealed a reduction in autophosphorylation intensity with increasing amounts of ΔN-TLK2-KD (Fig. 2c), suggesting that TLK2 autophosphorylation follows a unimolecular process and that autophosphorylation occurs in the context of the protein dimer.

While over 100 potential TLK substrates have been proposed[40], only a few have been validated or examined in any functional detail. These include ASF1, a histone H3-H4 chaperone, and RAD9, a member of the 9-1-1 complex that plays an important role in the DDR[17,18,29–31]. To test substrate phosphorylation by the various TLK2 constructs, we used the generic kinase substrate MBP and bacterially expressed ASF1a and monitored phosphorylation in a reaction triggered with radiolabelled ATP. Consistent with the autophosphorylation data, the Kdom variants were much less active in catalysing phosphate transfer to MBP compared to the ΔN-TLK2 constructs (Fig. 2d, Supplementary Figure 4). An increase in activity on MBP was observed in all of the phosphorylated variants when compared to their respective unphosphorylated forms. Minor differences were observed in the activity of the phosphorylated variants due to the presence or absence of the C-tail, which appeared to decrease activity in the context of the ΔN-TLK2 proteins, consistent with its proposed negative regulatory role in TLK1 (Fig. 2d)[25,27].

However, when the reaction was carried out with the specific substrate ASF1a, no activity was observed with the Kdom variants, suggesting that ASF1a recognition is dependent on the CC domains (Fig. 2e, Supplementary Figure 4) and that ASF1a is phosphorylated only by variants of TLK2 that are capable of oligomerization. It is also notable that while the phosphorylated ΔN-term TLK2 forms were much more active than the unphosphorylated forms on MBP, less of a difference was seen when ASF1a was used as a substrate. In the absence of the C-tail, proteins exhibited similar activity, regardless of whether they were co-expressed with lambda phosphatase, while in the presence of the C-tail, phosphorylation had an inhibitory effect (see Fig. 2e). These data suggested that TLK2 activity is more regulated in the case of the physiological substrate ASF1a.

**Mapping of TLK2 phosphosites**. Our experiments showed that all the variants were active when expressed in *E. coli*, suggesting that TLK2 does not need the action of exogenous kinases for activation. In addition, our biochemical characterization indicated that the monomeric kinase domain, although active, displayed a negligible catalytic activity compared to the dimeric form of the enzyme (Fig. 2b), which contains the CC region. Moreover, phosphorylated forms of the dimer displayed higher orders of oligomerization, while the kinase domain constructs remained monomeric. To understand the mechanism of TLK2 activation, we sought to identify the phosphorylation sites of TLK2 in our controlled system expressing the protein in *E. coli*. Using mass spectrometry (MS), we obtained full peptide coverage of the ΔN-TLK2 construct produced in the presence or absence of lambda phosphatase and the ΔN-TLK2-KD mutant. New phosphorylation sites were observed in our samples, especially in the 191–450 region around the CC domains (Fig. 3a–c, Supplementary Data 1). To determine if these phosphosites were also detected in

**Fig. 1** Architecture and characterization of TLK2. **a** TLK2 domain architecture and constructs used in this study. For a complete list see Supplementary Figure 2. **b** Size exclusion chromatography (SEC) coupled with Multi-Angle Laser Light Scattering (MALLS) profiles were used to assess the oligomerization state of the phosphorylated and unphosphorylated constructs with or without the C-tail for the kinase domain and **c** ΔN-TLK2 constructs. The effect of the catalytically inactive mutant ΔN-TLK-KD (D613A mutation) is also included. **d** The indicated TLK2 mutants were overexpressed in AD293 cells by transient transfection and pulled down from cell lysates using Streptavidin resin. Protein–protein interactions were analyzed by western blotting for TLK2, TLK1, ASF1 and LC8 (right panel). Input levels are shown in the left panel including the Ponceau red-stained membrane showing similar total protein levels. The KD in this experiment bears the D592V mutation instead of D613A. **e** Kinase assays were performed from Streptavidin pull-downs performed as in **d**. Kinase complexes were incubated with either MBP or ASF1a in the presence of $^{32}$P-ATP and autophosphorylation (TLK2) and substrate phosphorylation assessed in dried SDS-PAGE gels exposed to a phosphoimager. **f** Immunofluorescence analysis of exogenous TLK2 in transiently transfected AD293 cells. Localization of TLK2 and mutant forms was determined by staining with anti-FLAG antibodies and co-staining of nuclei with DAPI. Cytoplasmic localization of the ΔN-TLK2 mutant confirms the presence of the NLS in the N-terminus of the protein. The scale bar represents 10 μm in all the pictures. Uncropped gels and blots are shown in Supplementary Figures 10–15

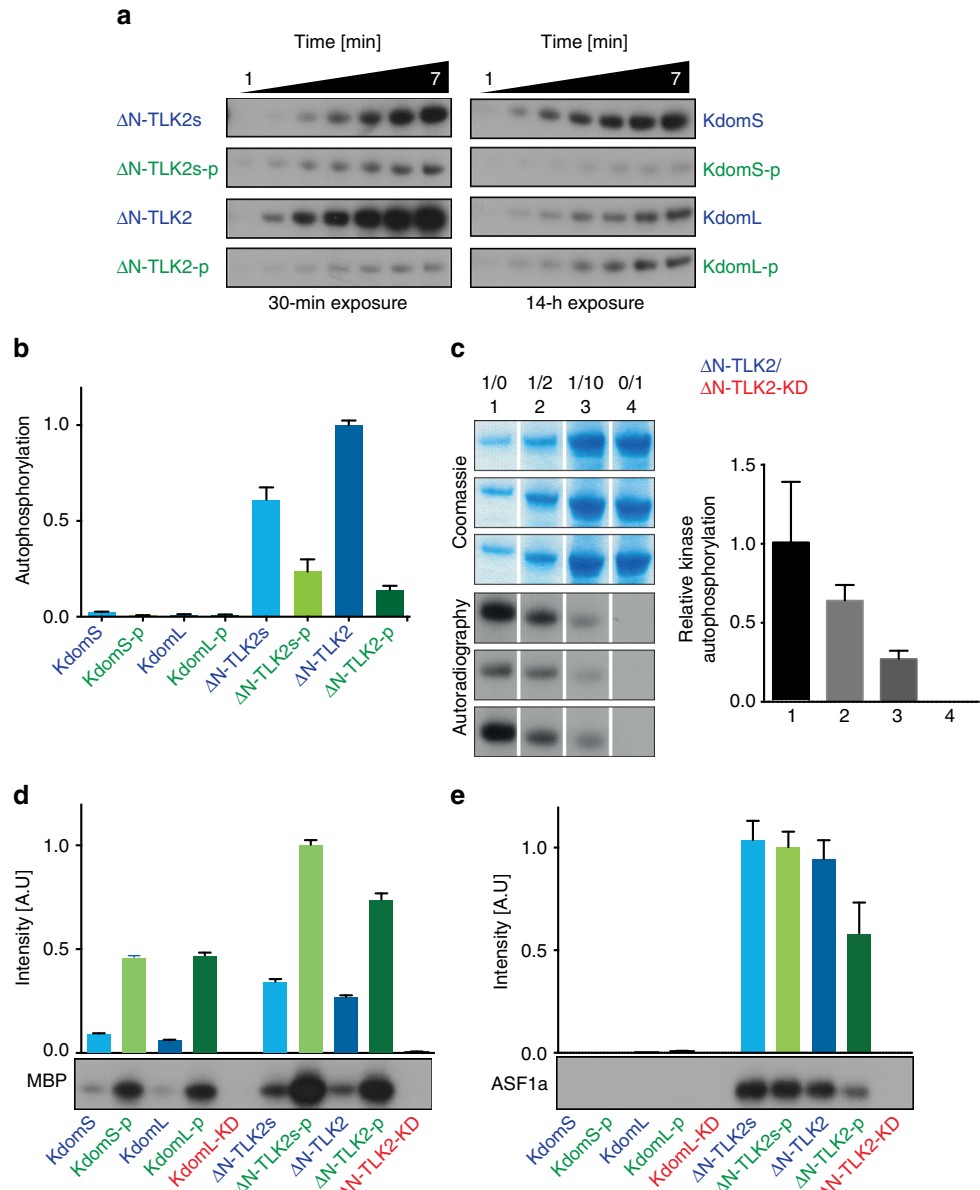

**Fig. 2** Catalytic activity depends on TLK2 autophosphorylation. **a** Autoradiograms of TLK2 autophosphorylation activity for the kinase domain and the ΔN-TLK2 constructs after a 30-min or 14-h exposure, respectively (see Supplementary Figure 4a for SDS-PAGE and the initial velocity plots). **b** Quantification of the autophosphorylation assay showing that the kinase domain is not efficient as the ΔN-TLK2 constructs. The unphosphorylated ΔN-TLK2 displays the larger activity in autophosphorylation. Data points indicate the relative autophosphorylation normalized to the autophosphorylation activity of ΔN-TLK2 (mean ± s.d., $n = 3$ biological replicates). See Supplementary Figure 4 for the details of the autophosphorylation rates of KdomL-p and ΔN-TLK2-p. **c** The autophosphorylation of ΔN-TLK2 is a unimolecular reaction. The increasing ratios of the ΔN-TLK2-KD inactive mutant diminished the activity of the kinase. The data points indicate the relative autophosphorylation of the ΔN-TLK2/ΔN-TLK2-KD reactions normalized to the activity of point 1 (mean ± s.d., $n = 3$ biological replicates). **d**, **e** Phosphorylation of TLK2 substrates. The activity is represented in a histogram for MBP and ASF1a phosphorylation by various TLK2 constructs. The data points indicate the relative substrate phosphorylation for the TLK2 constructs normalized to the TLK2 construct with the highest activity (mean ± s.d., $n = 3$ biological replicates). See Supplementary Figure 4 for autoradiograms and SDS-PAGE for MBP and ASF1a phosphorylation

mammalian cells and whether they were influenced by the predicted NLS in the N-terminal fragment (Fig. 1a), we produced ΔN-TLK2 and the full-length WT TLK2 in HEK293 cells.

We could detect the phosphosites in the 1–191 protein segment, which have been previously reported in PhosphoSitePlus. Minor differences were observed between ΔN-TLK2 and the full-length protein expressed in mammalian cells (Fig. 3c, Supplementary Data 1). The absence of some of the phosphosites in ΔN-TLK2 expressed in HEK293 may be due to the presence of Ser/Thr phosphatases in mammalian cells that could remove

them during the purification. Another possibility is that the large expression of ΔN-TLK2 in the prokaryotic system leads to phosphorylation of secondary sites with lower or no functional relevance, which are, therefore, not observed in mammalian cells. The differences of phosphorylation in the presence and absence of the 1–190 N-terminal fragment are scarce, suggesting that the nuclear localization of the enzyme does not change dramatically its phosphorylation pattern (Supplementary Figure 5a–c).

Therefore, ΔN-TLK2 expressed in *E. coli* displays a phosphosite landscape in similar regions to ΔN-TLK2 expressed in

HEK293 cells. We continued our biochemical analysis in the *E. coli* isolated sample which provides a more controlled scenario for our study.

Altogether, the homodimeric ΔN-TLK2 expressed in *E. coli* displays 41 phosphosites out of 67 possible Ser/Thr sites (Fig. 3c),

15 of which were in the kinase domain and six of them mapped onto the C-tail. The rest, except S330 in the predicted CC2, were found in the loops between the predicted CC domains. The heat map based on hierarchical clustering analysis and the volcano plot display the distribution of phosphorylation sites on TLK2,

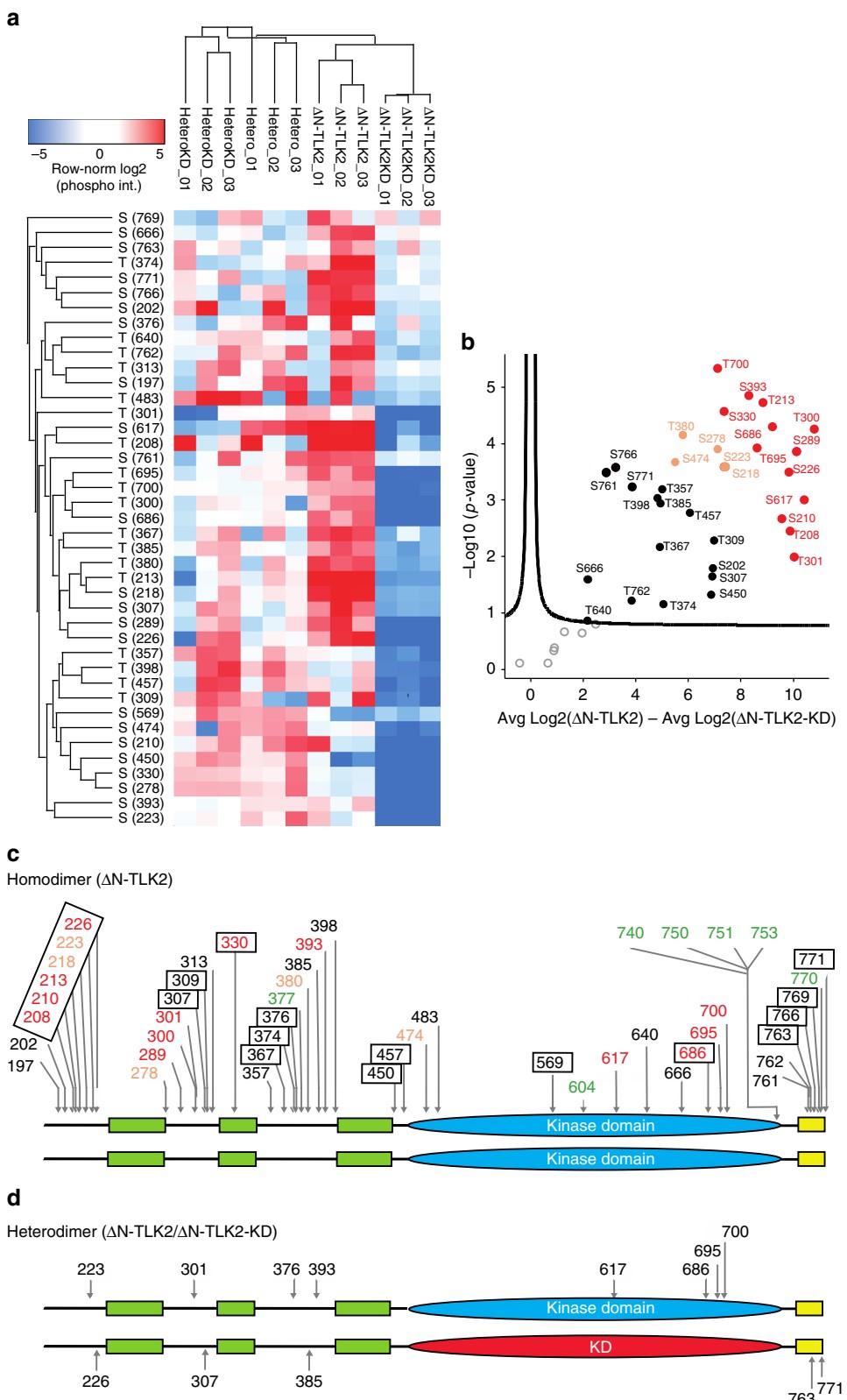

which clearly differs between active TLK2 and the KD variant (Fig. 3a, b).

To understand the role of phosphorylation in the activation process within the TLK2 dimer, we co-expressed in *E. coli*, in the absence of lambda phosphatase, differentially tagged constructs, His-ΔN-TLK2 and Strep-LSL-ΔN-TLK2-KD, in order to isolate a heterodimer where only one of the kinase domains was inactive (Fig. 3d, Supplementary Figure 6a). The presence of different tags and the absence of interconversion of the heterodimer (checked by MALLS after purification) allowed the MS analysis of the phosphopeptides from each distinct monomer. We obtained full coverage of both monomers (Supplementary Figure 6b–d and Methods) and the heat maps of the phosphosites shows two clear clusters between the homodimers and the heterodimers (Fig. 3a). A clear overall difference can be observed in the shared phosphosites and the intensity/levels of phosphorylation in the protein samples. To analyze specific differences, a *t*-test was performed and the corresponding volcano plots showed a difference in the phosphopeptide sites between the ΔN-TLK2 and ΔN-TLK2-KD homodimers (Supplementary Figure 6b). In addition, clear phosphorylation differences were also observed in the ΔN-TLK2-KD monomer depending on whether it was in the context of a heterodimer or a homodimer (Supplementary Figure 6a), while little differences were observed between the monomers of ΔN-TLK2 and ΔN-TLK2-KD in the context of the heterodimer (Fig. 3a, Supplementary Figure 6d).

The unique phosphosites observed in either the heterodimer active subunit or in the heterodimer KD subunit were mapped onto the molecule to illustrate the distinct phosphorylation sites on each monomer (Fig. 3d). As expected the heterodimer contained fewer phosphorylations in comparison to the wild-type homodimer, as only 5 and 8 unique sites were found in ΔN-TLK2-KD and ΔN-TLK2 monomers, respectively (Fig. 3d). We observed that the heterodimer showed unique sites between the two monomers, in agreement with our competition experiment (Fig. 2c). Therefore, in this heterodimeric sample, the phosphorylations must arise from the active monomer. These unique phosphosites can be categorized into two groups, a first one including the phosphosites in the ΔN-TLK2-KD resulting from inter-molecular phosphorylation (*trans*), and a second group arising from intra-molecular phosphorylation (*cis*) in the ΔN-TLK2 (Fig. 3d). The active ΔN-TLK2 monomer of the heterodimer undergoes *cis* phosphorylations (S617, S686, T695 and T700) in the kinase domain and (S223, T301, S376 and S393) in the oligomerization domain (Fig. 3d). The *trans*-phosphorylations in the ΔN-TLK2-KD monomer occur only in the C-tail (S763 and S771) and the oligomerization domain (S226, S307 and T385). A comparison of these results revealed that the heterodimer lacks 17 phosphosites that are present in the active homodimer. This indicated that the activity of both kinase domains in the dimer is needed to achieve full

autophosphorylation of the molecule and suggests that initial phosphorylations by one of the kinase domains in the dimer makes new sites available that would be subsequently targeted by the second catalytic domain. Therefore, we hypothesize that initial autophosphorylation between the monomers could trigger conformational changes leading to the full activation of the enzyme. One example of such a possible stepwise autophosphorylation can be observed in the C-tail where S771 and S763 are due to *trans*-phosphorylations (Fig. 3d), while the other phosphosites in the C-tail (Fig. 3c) could occur in *cis* after the phosphorylation of these residues, as they do not appear in the heterodimer (Fig. 3d). Whether the TLK2 dimer is parallel or antiparallel cannot be determined at this stage. However, a large number of the phosphosites in the CC region, which have been also detected in the protein expressed in mammalian cells, seem to arise from *trans*-phosphorylation, suggesting that the CC region of a monomer could be located in the proximity of the kinase domain of the other subunit of the dimer.

**Structure of the TLK2 kinase domain**. TLKs represent a unique kinase family, defined by divergent protein sequence, mainly in the activation loop, that is located between the Polo and AGC kinase families in the kinome[41]. The catalytic domain of the protein kinases shares a common fold consisting of an N-terminal and a C-terminal lobe[42,43]. Two hydrophobic interaction networks, termed 'R- (regulatory) and C- (catalytic) spines', cross the two lobes (Supplementary Figure 7). These elements are essential for catalysis and together with the central F-helix[44,45] and encompass the activation loop, the catalytic loop and the DFG motif, which constitute the kinase core.

Here, we present the structure of the kinase domain of human TLK2; the unphosphorylated TLK2 kinase domain was crystallized in complex with ATPγS[46], a slowly hydrolysable ATP analogue, and the structure solved at 2.8 Å resolution (Fig. 4a, data collection and refinement statistics are shown in Table 1). The divergent sequence conservation of the catalytic domain of TLK2, when compared to other kinases, is reflected in its structure. A BLAST search of protein homologues in the PDB and a subsequent sequence alignment and structural superposition, as implemented in ENDSCRIPT[47], showed major structural differences. The sausage plot indicates low sequence identity and large r.m.s. deviations in many regions of the kinase domain, especially those involving key regulatory elements, such as the activation loop (Fig. 4b, c, Supplementary Figures 7–8).

Our unphosphorylated ATPγS TLK2 structure is in a pre-active conformation. Although the R-spine is in the correct alignment to create a platform for an active conformation, important hallmarks of the active state, such as the salt bridge between the conserved lysine and glutamate in the β3 and αC helix (K491 and E514 in TLK2) are not observed (Fig. 4c). The main characteristic features of TLK2 are its unusual ATP-binding

**Fig. 3** TLK2 undergoes both *cis*- and *trans*-phosphorylation in the dimer. **a** Hierarchical clustering analysis of the phosphorylation sites of the ΔN-TLK2 constructs expressed in *E. coli* displayed in a heat map. Log-transformed and row-normalized intensities of phosphosites are shown in triplicates for the heterodimer kinase-dead subunit (HeteroKD), the heterodimer active subunit (Hetero), the active homodimer (ΔN-TLK2) and the kinase-dead homodimer (ΔN-TLK2-KD). **b** Volcano plot validation showing the regulation and significance of phosphosites between the ΔN-TLK2 homodimer and the kinase-dead ΔN-TLK2-KD homodimer. Phosphosites are labelled in black (significant), light orange (highly significant) and red (most highly significant). **c** All detected phosphorylation sites mapped on the ΔN-TLK2 domain scheme. The drawing does not imply a parallel arrangement of the dimer. Sites cannot be assigned to individual molecules of the dimer. Boxed phosphosites have been detected in both HEK293 and *E. coli*-expressed ΔN-TLK2, while unboxed sites were only found in the *E. coli*-expressed ΔN-TLK2. Font colour represents significance as shown in Fig. 3b for the *E. coli*-expressed ΔN-TLK2 (black, light orange and red). The unboxed green sites represent phosphosites observed exclusively in the ΔN-TLK2 expressed in HEK293 cells. **d** Schematic representation depicting the unique phosphorylation sites observed in the heterodimer ΔN-TLK2 active subunit and in the ΔN-TLK2-KD kinase-dead subunit. The phosphorylation sites in T208, T213, S218, S226, S289, T300, S307, T357, T380, S761 and T762 were observed in both subunits of the heterodimer (see Supplementary Data 1 and Fig. 6a–d for volcano plots displaying distributions of peptides when comparing the different constructs)

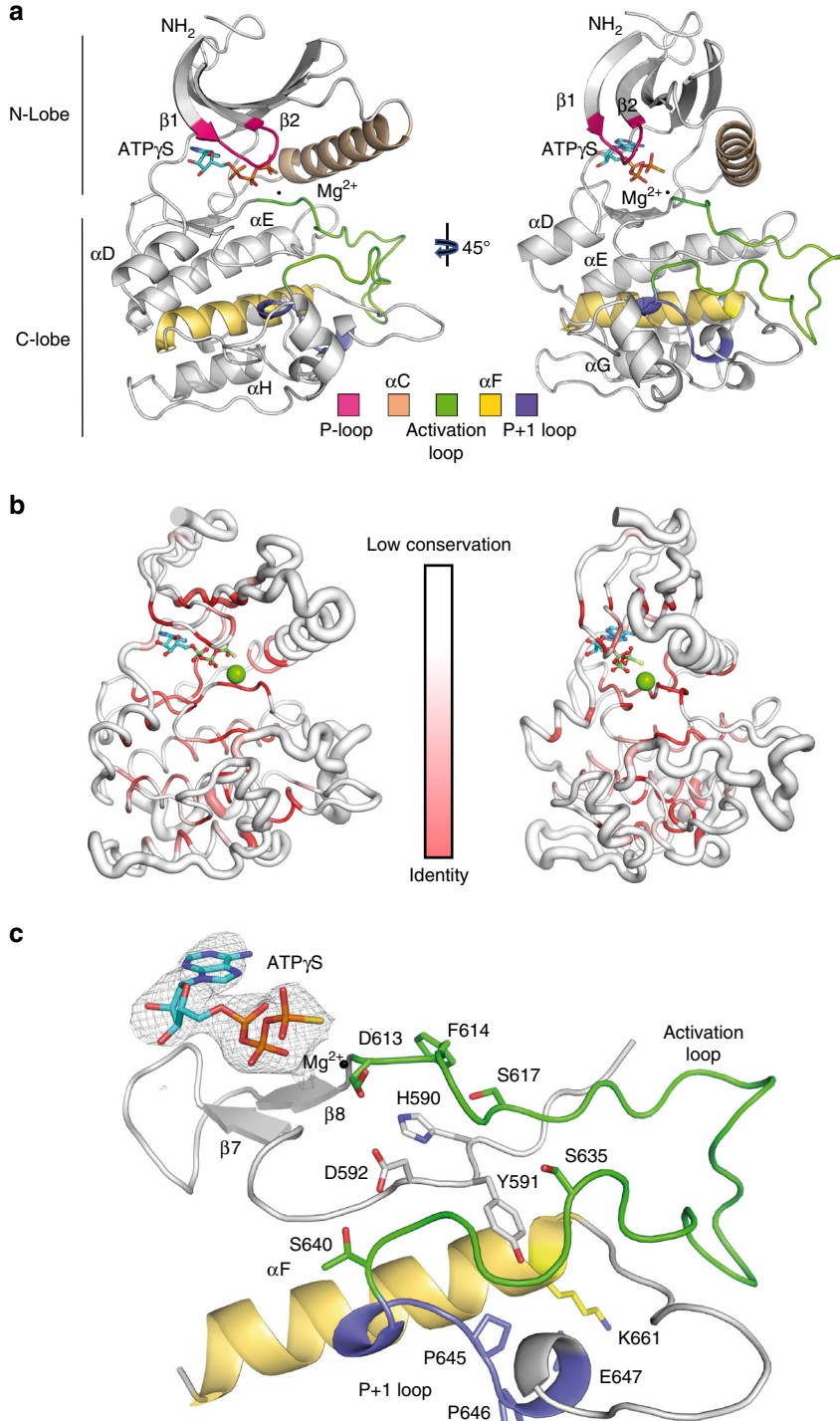

**Fig. 4** Crystal structure of TLK2 kinase domain. **a** Ribbon diagram of the TLK2 kinase domain structure in complex with ATPγS. The protein fold shows the classical N- and C-lobes and the important regulatory features associated with the kinase domain. The singular activation loop of TLK2 can be fully traced in the structure (see Table 1 for data collection and refinement). **b** To correlate structural and sequence conservation, we generated a sausage plot, showing a variable tube representation of the Cα trace after a PDB query for performed with ENDSCRIPT. For this drawing, homologous protein structures (1291 PDB files) were superposed onto the TLK2 kinase domain structure with ProFit. The size of the tube is proportional to the mean r.m.s. deviation per residue between Cα pairs. The white to red colour ramping is used to visualize sequence conservation. **c** A detailed view of the structure showing the ATPγS, catalytic and activation loops, P+1 loop and the αF helix. The key regulatory residues such as the DFG loop, the HYD, E647 and K661 are depicted in sticks. An omit map for the ATPγS molecule at 1.25 σ level is included in the figure

**Table 1 Crystal structure of TLK2 kinase domain, data collection and refinement statistics**

| Data collection | TLK2 crystal, PDB: 5O0Y |
|---|---|
| Wavelength (Å) | 1 |
| Resolution range (Å) | 72.76–2.86 (2.963–2.86) |
| Space group | $P\,2_1 3$ |
| Unit cell (Å), ° | $a = b = c = 126.02;\ \alpha = \beta = \gamma = 90$ |
| Total reflections | 31,396 (3094) |
| Unique reflections | 15,698 (1547) |
| Multiplicity | 2.0 (2.0) |
| Completeness (%) | 99.97 (100) |
| Mean $I$/sigma ($I$) | 13.12 (0.54) |
| Wilson B-factor | 97.07 |
| R-merge | 0.032 (1.29) |
| R-mean | 0.045 (1.82) |
| CC1/2 | 0.999 (0.26) |
| CC[a] | 1 (0.64) |
| **Refinement** | |
| Reflections used in refinement | 15,694 (1547) |
| Reflections used for R-free | 790 (76) |
| R-work | 0.20 (0.36) |
| R-free | 0.22 (0.35) |
| Number of non-hydrogen atoms | 2390 |
| Macromolecules | 2359 |
| Ligands | 31 |
| Protein residues | 288 |
| RMS (bonds) (Å) | 0.01 |
| RMS (angles) ° | 1.64 |
| Ramachandran favoured (%) | 93.1 |
| Ramachandran allowed (%) | 5.9 |
| Ramachandran outliers (%) | 1.0 |
| Rotamer outliers (%) | 3.8 |
| Clashscore | 2.5 |
| Average B-factor | 104.84 |
| Macromolecules | 86.33 |
| Ligands | 125.26 |
| Number of TLS groups | 1 |

Statistics for the highest-resolution shell are shown in parentheses
[a]Friedel mates were averaged when calculating reflection statistics

motif, GxGxxS (position 469–474), also observed in CK2a and Cdc7, instead of the canonical GxGxxG seen in other kinases[42], and a long activation segment. This element is well-defined in the structure, exhibiting a stable conformation (Fig. 4c). Within the activation segment is the T-loop, which is located in the C-terminal part of the extended activation segment, and whose phosphorylation is considered a critical regulatory element in protein kinases[48]. Although there are some exceptions, such as CaMK-II, many kinases contain a His–Arg–Asp (HRD or RD) motif within the catalytic loop and the positive charge on the Arg side chain serves to stabilize the phosphorylated activation loop site[49]. However, TLK2 does not contain this motif, a Tyr residue is located in the Arg position, suggesting that TLKs may not need T-loop phosphorylation for activation (Fig. 4c, Supplementary Figure 1).

All the phosphorylation sites detected in the kinase domain are shown in stick representation (Fig. 5a). Out of the 15 phosphosites, we can locate nine of them in the structure of the kinase domain, mainly in the C-lobe, around the catalytic and activation loops. Unfortunately, we cannot observe the residues phosphorylated in the C-tail (T750 to S771) because this region was not included in our crystallized protein. Our experiments suggest that *cis* autophosphorylation is likely to be responsible for phosphorylating the S617, S686, T695 and S700 sites, while S474, T483, S569, T640, and S666 could result from *trans*-

autophosphorylation within the dimer (Fig. 3e). Phosphorylations in the P-loop have been observed to influence the regulation of other kinases. The modelling of a phosphoserine instead of S474 in the P-loop of the kinase domain (Fig. 4a, Fig. 5a) suggests that this modification could affect nucleotide binding, leading to a reduction in activity. There are few examples of P-loop phosphorylation. One is found in Bcr–Abl, where pY272 impairs binding to STI-571, resulting in a lack of response to the drug. Additional examples are pS150 of the bacterial Ser/Thr kinase HipA[50] and pY15 of CDK2[51]. Phosphorylation in these residues has been shown to inhibit kinase activity by disturbing ATP binding or by blocking the substrate-binding site, respectively. The other observed phosphorylation in the TLK2 N-lobe, pT483, could interact with the loop preceding the initial β-strand of the N-lobe, rigidifying the connection to the CC3. This could contribute to stabilizing the kinase domain and avoiding flexibility of this moiety. The rest of the phosphoresidues are present in the C-lobe. pS569 is located near the C-spine and its phosphorylation will likely aid in fixing the conformation of the catalytic loop through interactions with the main chain of A605 and C606. Its mutation to alanine would not allow this interaction, but it could be favoured by hydrophobic contacts with A605, thus not affecting the activity (Fig. 5a, b, Supplementary Figure 9a, b). Although it has not been observed in our mammalian expressed protein, an interesting phosphorylation occurs in pS617, which is located just after the DFG motif. A phospho-mimic mutation (Asp) of this residue inhibited the kinase activity (Fig. 5a, b, Supplementary Figure 9a, b). This is likely due to the fact that the presence of a phosphate will disturb the F614 conformation, affecting the R-spine. A residue conserved in both TLK1 and TLK2, T635, is in the position analogous to the activating phosphorylation of CDK2 and this residue was found phosphorylated in TLK1 (Anja Groth, personal communication). However, as previously mentioned, TLK1 and TLK2 do not contain an RD motif, suggesting that this would not be required for activation. We have found only unphosphorylated peptides containing this residue in our MS experiments both in *E. coli* and HEK293. Therefore, its phosphorylation does not seem to be essential for activity in vitro or likely to be the result of *cis* or *trans*-autophosphorylation. The S635A and S635D mutations in this residue reduced the activity of TLK2 but had converse effects on overall autophosphorylation. While S635A showed somewhat reduced autophosphorylation compared to TLK2-WT, it was increased by the S635D mutation. This could indicate that S635 is modified by another kinase in vivo to modulate TLK1/2 activity. Near T635, pT640 could further stabilize the catalytic loop by interacting with the main chain of the 590-HYD-592 motif, holding an active catalytic loop conformation together with T640. Interestingly, pT686 and pS695 seem to rigidify the loops connecting the αG-helix most likely favouring the kinase activity by stabilizing the P+1 loop region. The alanine mutations of these two residues decrease the activity of the enzyme. Regarding pS700, this residue is exposed to the solvent and is not clear how its modification could affect the kinase activity.

**Intellectual disability mutations impair TLK2 activity.** A recent meta-analysis of data from over 2000 patients identified TLK2 as one of ten new candidate genes for ID and other neurodevelopmental disorders[38]. These patients have de novo loss of function mutations (DNM) and exhibit severe clinical features, such as facial dysmorphisms. Two of the DNM mutations in TLK2 were located on the N-lobe around the ATP-binding pocket, one was found in the activation loop and the last one in the C-lobe (Fig. 5c). Amino acids substituted on the kinase domain are highlighted in sticks and surface representation to show the

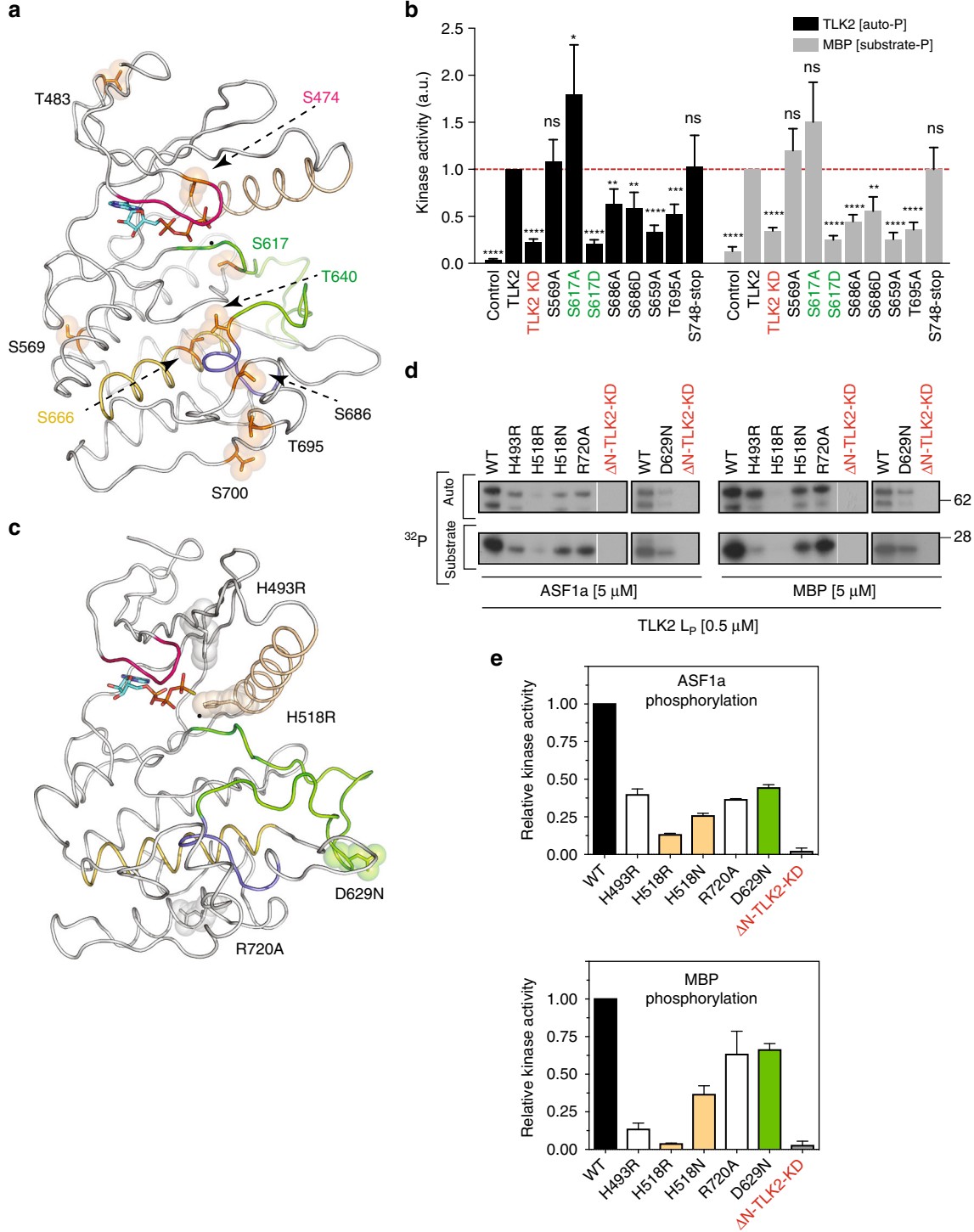

**Fig. 5** Phosphorylation sites in the kinase domain. **a** Nine out of the 15 phosphorylation sites in the kinase domain can be modelled on the structure. The sites are highlighted in stick and sphere representation and lie in the N-lobe, activation loop and C-lobe. The six phosphosites on the C-tail cannot be observed because the crystallized construct lack that segment. **b** In vitro kinase assay of the indicated phosphorylation mutants of TLK2 isolated from AD293 cells by Streptavidin pull-downs showing relative autophosphorylation and MBP phosphorylation (see Supplementary Figure 9a for the autoradiograms and Supplementary Figure 9b IP pull-downs). The KD in this experiment bears the D592V mutation instead of the D613A. **c** Structure of the kinase domain highlighting the position of the intellectual disorder and cancer mutants. The amino acids where mutations have been reported are shown in stick–sphere representation. **d** In vitro kinase assay of the different ID and cancer mutants using MBP and ASF1a as substrates. **e** Quantification and comparison with the wild-type protein is represented in a histogram. The data points indicate the relative kinase activity of the TLK2-mutants normalized to the activity of the wild-type protein (mean ± s.d., $n = 3$ biological replicates). See Supplementary Figure 9c–f for MBP and ASF1a phosphorylation and autophosphorylation activities by the mutants

location of these residues with respect to the activation segment (Fig. 5c). Many of the substitutions are far away from the catalytic loop. Nonetheless, we decided to investigate the activity of these de novo mutations (H493R, H518R, D629N, R720A) by producing recombinant phosphorylated ΔN-TLK2-p mutants and measuring their ability to phosphorylate MBP and ASF1a. Kinase assays revealed H518R to be most severe, resulting in 85–90% reduction of kinase activity (Fig. 5d, e). The H518R mutation is located on the bottom of αC-helix and is conserved in other kinases, constituting the RS3 of the regulatory spine[44] (Supplementary Figure 7). Mutations resulting in larger side chains, such as an arginine, will disturb the organization of this key element and therefore a decrease in its activity was expected. We generated a more conservative mutation, H518N. This protein showed a 50% increase in activity on ASF1a compared to H518R, but nonetheless, H518N was only 25% as active as the wild-type protein, supporting our hypothesis (Fig. 5d). The H493R mutation showed a 65% reduction in kinase activity compared to the wild-type TLK2 (Fig. 5d, e, Supplementary Figure 9c, d). The side chain of H493 also faces the ATP-binding pocket and is close to the Glu–Lys salt bridge, so a bulkier side will affect the kinase activity by distorting the catalytic side. While all of these mutants are located in the N-lobe next to the catalytic site, the loss in activity by R720A was the most surprising. This residue is located on the penultimate helix (αH) with the side chain interacting with D738. Therefore, its mutation to alanine will disrupt this interaction. One possibility is that the disruption of this interaction could induce a reduction in activity by destabilizing the C-tail positioning, thus impairing its autophosphorylation by the enzyme. The larger decrease of kinase activity reduction observed for the D629N mutation in the case of ASF1a in comparison with MBP suggested that its effect could be related to specific substrate recognition. This residue is not conserved in other kinases and is exclusive to the TLKs that are defined in part by their long and divergent activation loop. We also compared the autophosphorylation capability of these mutants with the wild-type protein (Supplementary Figure 9e, f). They showed a more than 50% decrease in autophosphorylation, thus following a similar trend in activity on ASF1a, supporting a mechanism by which autophosphorylation triggers TLK2 to an active conformation for substrate recognition and catalysis.

**Docking of small-molecule inhibitors of TLK2 activity.** The implication of TLK2 in some types of cancer has suggested that small-molecule inhibitors of its activity could be used as anticancer agents[36]. While specific inhibitors of TLK2 have not yet been identified, we tested several compounds previously identified to have activity against TLK2 in a large screen of commercially available small molecules[52]. Using immuno-precipitated protein overexpressed in AD293 cells, we found that the non-specific kinase inhibitors Staurosporine and Nocardiopsis completely abolished TLK2 catalytic activity (Fig. 6a). We then tested several inhibitors showing activity towards TLK2 that are known to target CDK1 (CGP74541A) and GSK3 (Inhibitor XIII), as well as the Indirubin derivatives E804 and indirubin-3′-monoxime (Fig. 6a). Indirubin is an active component of Danggui Longhui Wan, a traditional Chinese medicine formulation, whose encouraging clinical results in chronic myelocytic leukaemia patients have stimulated numerous studies on this compound[53]. With the exception of the monoxime variant of indirubin, the other three compounds substantially inhibited TLK2 kinase activity (Fig. 6a). Using our structure, we performed an initial analysis modelling the chemical compounds in our kinase domain structure using HADDOCK[54] in order to rationalize the molecular bases of the observed inhibition (Fig. 6b, c). A future

in-depth study including more compounds will be needed to evaluate possible inhibitors. In our study, the top 40 docking solutions were selected for each compound and examined in detail. Out of them, the best docking solutions without steric clashes with the protein are shown (see Methods). In the case of CGP74541, two solutions with minor differences were found, while for indirubin E804, only one solution did not show clashes with the protein. In the case of the GSK3 inhibitor XIII, four different solutions could be docked by HADDOCK in the nucleotide-binding pocket of the kinase domain. While these compounds showed a substantial reduction in TLK2 activity, the indirubin-3′-monoxime did not affect TLK2 activity. Structural modelling indicated that this different behaviour could be due to the absence of the aliphatic chain in 3′ and the presence of a bulky iodine in the indol ring. The aliphatic region is engaged in interactions in the cavity while the iodine atom will clash with the protein moiety, thus avoiding binding and inhibition. These preliminary results show that the structure of the kinase domain of TLK2 can be successfully used in virtual screening for TLK2 inhibitors.

## Discussion

TLKs have been implicated in key cellular processes ranging from chromatin assembly to DNA replication, DNA repair and transcription[18,23,24,55,56]. While genetic analyses in mice and cells suggest largely redundant roles for TLK1 and TLK2 in development and genome maintenance[26], mutations specifically in TLK2 were identified in patients with neurological disorders and ID[38], potentially reflecting the critical role in placental development that we recently reported, and specific functions have been proposed for TLK1 and TLK2[32]. In addition, frequent TLK2 amplifications were identified in oestrogen receptor (ER)-positive luminal breast cancers and TLK2 inhibition, alone or in combination with tamoxifen, substantially inhibited the growth of MCF7 xenograft tumours[36]. These data provide evidence for the therapeutic value of TLK2 inhibition in breast cancers, where ASF1b levels have been reported to also be elevated[57], and potentially other types of malignancies.

Although TLK activity has been shown to play an important role in the regulation of chromatin assembly, and potentially other processes, little mechanistic insight has been reported regarding the regulation of its activity. Our structure–function analysis of TLK2 reported here provides insights into the molecular basis of its activation and provides a molecular context for understanding this kinase family and the impact of mutations that have been identified in ID patients. The structure of the kinase domain will also facilitate the rational design of inhibitors that may be valuable for the treatment of cancer progression and the further interrogation of TLK function in model systems.

Our biochemical data indicate that the dimerization and further oligomerization of TLK2 plays an important role in its activation. TLK2 readily forms a dimer, and this dimeric arrangement appears to be scaled-up in the form of higher order oligomers following autophosphorylation (Fig. 2c), suggesting that aside from triggering activation, further oligomerization will increase the enzymatic activity by concentrating additional TLK2 molecules. TLK2 pull-downs show that it can associate with TLK1 and that the CC1 is an essential region for that interaction (Fig. 1d). Therefore, heterodimerization could be another regulatory layer in human TLKs, as it has been also observed in receptor tyrosine kinases (RTKs)[58].

Our activity assays show that the oligomeric arrangement of the enzyme is also crucial for activity and substrate phosphorylation. Autophosphorylation and therefore enzyme activation is rapidly achieved in ΔN-TLK2, while almost negligible in the

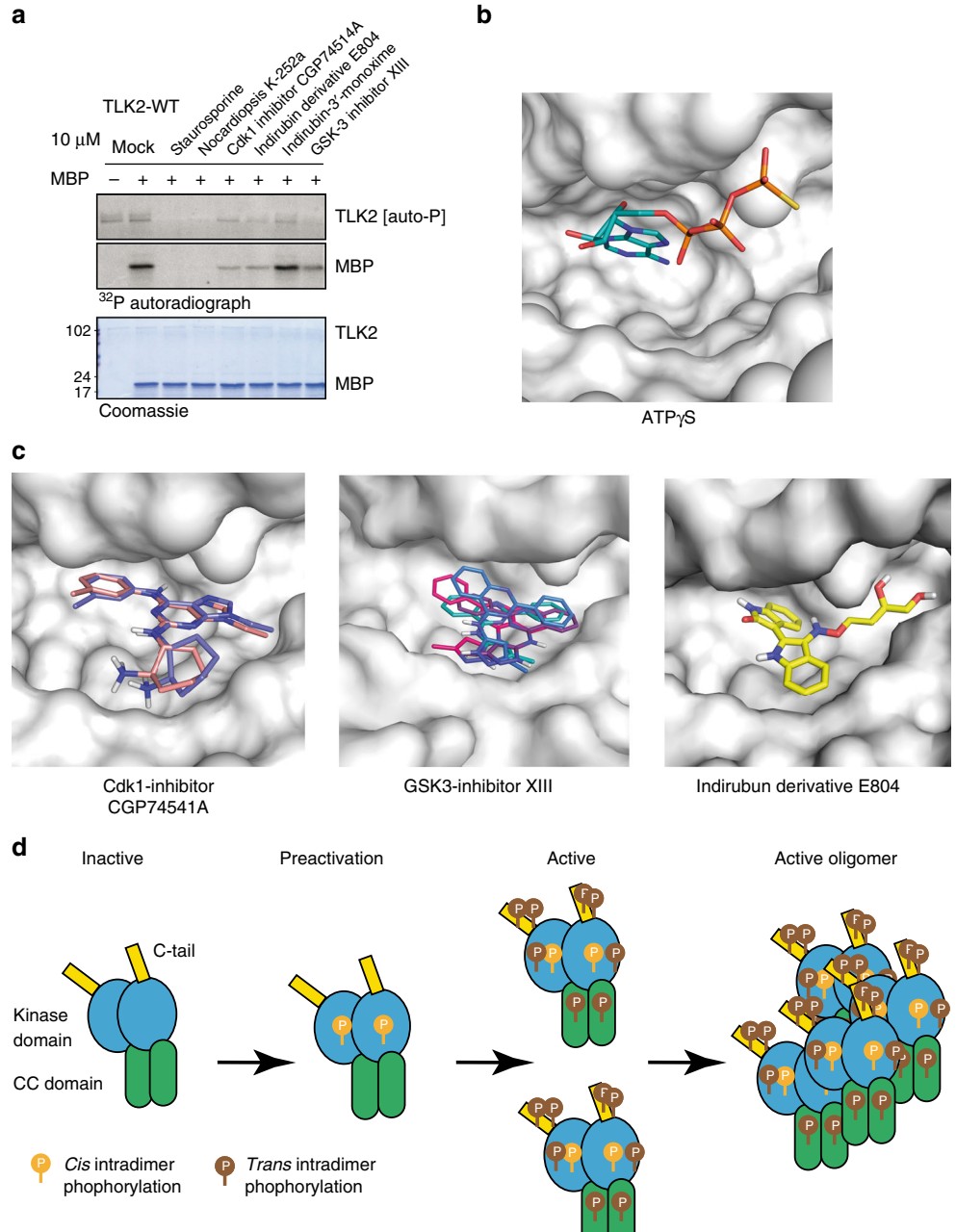

**Fig. 6** TLK2 inhibition and activation model. **a** In vitro kinase assay of TLK2 isolated from AD293 cells by Streptavidin pull-downs in the presence or absence of the substrate MBP and the indicated kinase inhibitors. All inhibitors were used at 10 μM and the Coomassie stained gel shows equal loading of TLK2 and MBP. **b** Detailed view of the ATP-binding pocket in the TLK2 kinase structure showing the bound ATPγS molecule. **c** Zoom of the ATP-binding pocket containing the modelled molecules shown to inhibit TLK2 kinase activity. The best score modelling results displayed no steric clashes. **d** Model of TLK2 activation

kinase domain constructs (Fig. 2a, b), which is in agreement with the unimolecular behaviour of the autophosphorylation reaction (Fig. 2c). Indeed, none of the phosphosites in the heterodimer containing a catalytically impaired kinase domain is shared between the two monomers (Fig. 3d), supporting the hypothesis that autophosphorylation occurs inside the dimeric arrangement. Moreover, dimeric constructs are capable of phosphorylating ASF1a while the kinase domains cannot, indicating that the CC segment is involved in substrate recognition (Fig. 2d, e).

Our MS analysis showed that 41 out of 67 possible Ser/Thr residues in ΔN-TLK2 expressed in *E. coli* are phosphorylated in

our assay. Twenty-six of these sites are located in the CC region, mainly in the loops joining them, 15 are found in the kinase domain, six of which lie in the C-tail (Fig. 3c). The presence of a substantial fraction of these sites in the CC region has been also observed when ΔN-TLK2 was expressed in HEK293 cells (Supplementary Figure 3d, Supplementary Data 1) This represents the first identification of functional autophosphorylation sites in TLK2. Thus far, the majority of the phosphorylation sites reported in public data are found in the N-terminal region and the C-tail domain, that in TLK1 has been shown to be the site of regulatory phosphorylation by CHK1[25,27,59]. S563 in TLK1 that corresponds to S569 in TLK2, which we identified specifically in

the context of the active homodimer, represents the only phosphorylation site in the kinase domain thus far reported[59]. In our study, we found that 11 of the 25 new sites that we identified in the CC region are conserved in TLK1. Many of them have also been found when TLK2 and ΔN-TLK2 were expressed in HEK293. While our data between bacteria and human cells are largely consistent, we do not pretend that every site occurs under physiological conditions. Here, we provide this information and test four of them (Fig. 5b, Supplementary Figure 9), but they will require further functional analysis in future work. In addition, more than 20 phosphorylation sites have been reported in the N-terminal domain of TLK2, suggesting that this is potentially an important regulatory domain in vivo[59]. Compared to the wild-type protein, we have observed consistently higher protein levels of an N-terminal mutant expressed in cells and this protein shows increased activity, suggesting a role in negative regulation (Figs. 1d, e). Nevertheless, this observation does not seem to be due to a difference in the phosphorylation in the absence of the N-terminal fragment (Supplementary Figure 5a, b).

The analysis of the phosphorylation sites observed in the heterodimer showed a large decrease in the number of phosphoresidues, which also displayed an asymmetric distribution. This suggests that a sequence of *cis* and *trans*-phosphorylations between the monomers in different regions of the molecule may occur to facilitate the activation of the enzyme. At least eight phosphorylations appear in *cis* (S223, T301, S376, S393, S617, S686, S695 and T700) while five are due to *trans*-phosphorylations (S226, S307, T385, S763 and S771). A detailed analysis of the phosphosites in the kinase domain showed that pS763 and pS771 in the C-tail, that have also been observed in HEK293 expressed TLK2, are generated by the active monomer suggesting that these residues need to be trans-phosphorylated to allow the subsequent *cis*- or *trans*-phosphorylation of the other four residues in the C-tail (S761, T762, S766 and S769) and the rest of the kinase domain (Fig. 3e). Interestingly, a clear loss of phosphorylation sites in the heterodimer was observed in the loops of the CC region when compared with the homodimer (Fig. 3c, d). The absence of these phosphosites in the heterodimer could reflect the inability to fully activate the enzyme or be due to the fact that it cannot assemble into higher order oligomers because of the lack of phosphorylations in the C-tails. The absence of the C-tail in the ΔN-TLK2-p inhibits the assembly of large oligomers (Fig. 2c) indicating it is required to facilitate their formation.

Altogether, these data suggest a possible activation model where the *cis*-autophosphorylations in the kinase domain (S686, T695 and T700) would trigger a cascade of conformational changes that would allow the *trans*- and *cis*-phosphorylation of sites in the C-tail, as it has been described for members of the closely related AGC kinase family[60]. Notably, pS686 was also detected in the HEK293 expressed proteins. The fact that the monomeric kinase domains used in this study also displayed catalytic activity, albeit significantly reduced compared to ΔN-TLK2, indicated that the *cis*-phosphorylations occur in the monomeric kinase domain (Supplementary Data 1), but it cannot reach full activation due to their monomeric status. The fully activated TLK2 dimer would induce the formation of larger oligomers where the phosphosites in the loops joining the CCs could be further phosphorylated. The role of oligomerization in kinase activation has been also postulated for RTKs[58], IKKs[61] or the oncogenic fusion protein Bcr-Abl[62]. In this context, pT617, which we have shown reduces the kinase activity, and pS474 could play an important regulatory role in avoiding the phosphorylation of inappropriate residues reducing the activity of the enzyme (Fig. 6).

Our results contribute to the understanding of the molecular mechanism by which the TLK family kinases are regulated and provide important structural information that can facilitate the future design of inhibitors for use in cancer therapy, as well as information for the generation of phospho-specific antibodies to track kinase activity in cells or tissues. Moreover, our data provide insight into the consequences of the mutations observed in ID patients and suggest that further investigation into the regulation and dimeric structure of the TLKs is warranted.

## Methods

**Bacterial expression and purification.** For bacterial expression, TLK2 was cloned into MCS1 of the pET-Duet1 vector with an N-terminal Hexa-His tag either by restriction enzyme digest or In-Fusion cloning (Primer sequences and cloning strategy are shown in Supplementary Table 1, 2). To generate the dephosphorylated forms of the proteins, the lambda protein phosphatase was cloned into MCS2. Quickchange site-directed mutagenesis (Stratagene) was used to produce the KD mutants. The generate the heterodimer, Strep2-LSL-TLK2L-D613A was cloned into MCS2 of pETDuet containing His TLK2L in MCS1. All constructs were verified by DNA sequencing.

All constructs were expressed in either *E. coli* NiCo21(DE3) (NEB) or Rosetta (DE3) pLysS (Novagen, Millipore) at 37 °C. Expression was induced by the addition of 0.3 mM isopropyl-β-D thiogalactopyranoside (IPTG) to a mid-log culture, and then cells were harvested 3 h after induction. Recombinant proteins were purified from clarified crude cell extracts using a combination of immobilized metal ion affinity, anion exchange, Heparin and gel filtration chromatography (Purification buffer compositions are shown in Supplementary Table 3). All the TLK2-Kdom constructs were purified in two steps, NTA Ni-column chromatography followed by size-exclusion chromatography (Superdex 200, GE Healthcare). The longer forms of the protein required additional steps, either an extra monoQ for the dephosphorylated forms or a monoQ and a heparin for the phosphorylated forms, prior to SEC. Furthermore, unlike the dephosphorylated proteins, the phosphorylated forms behaved differently and bound to the monoQ column. SEC was performed in 20 mM Tris (pH 8), 150 mM NaCl and 0.25 mM TCEP. The peaks containing TLK2 were concentrated, and aliquots of pure proteins were flash-frozen in liquid nitrogen and stored at −80 °C. Protein purity was monitored by SDS–polyacrylamide gel electrophoresis (SDS–PAGE) and electrospray ionization MS. The list of constructs used in this study is shown in Supplementary Figure 2.

**Protein structure determination and refinement.** Crystallization of the kinase domain of TLK2 is described previously[46]. Briefly, crystals were only obtained when mixed with ATP-γ-S (2 mM) and appeared in two different screening conditions: Natrix condition No. 1 (crystal type *A*; 20 mM HEPES pH 7, 2 M Li₂SO₄, 10 mM MgCl₂) or Wizard II condition No. 37 (crystal type *B*; 1 M sodium/potassium tartrate, 0.1 M Tris pH 7, 200 mM Li₂SO₄), at 277 K. Larger crystals were reproduced in 24-well hanging-drop (Hampton Research Linbro plates) by mixing 1 μl protein and 1 μl reservoir solution. Crystals were mounted on CryoLoops (Hampton Research) and flash-cooled in liquid nitrogen. For data collection under cryogenic conditions, crystals were briefly soaked in a universal cryosolution, consisting of mother liquor supplemented with 20% (w/v) glycerol. The structure of the TLK2 kinase domain was determined by molecular replacement using Balbes[63]. Native datasets were collected from single frozen crystals at 100 K using a PILATUS detector at the PXI-XS06 beamline (SLS Villigen, Switzerland). Data processing and scaling were accomplished with XDS[64]. The initial molecular replacement solution was refined using PHENIX[65] and model building was performed with Coot[66]. Data collection and refinement statistics are summarized in Table 1. The identification and analysis of the hydrogen bonds and van der Waals contacts was performed with the Protein Interfaces, Surfaces and Assemblies service (PISA) and LIGPLOT at the European Bioinformatics Institute (http://www.ebi.ac.uk/msdsrv/prot_int/pistart.html). Figures were generated using PyMOL.

**Cloning of mammalian TLK2 constructs.** All TLK2 constructs expressed in mammalian cells for Fig. 1 were made in a pcDNA3.1 vector with an N-terminal Strep-FLAG (SF) tag (a kind gift from C.J. Gloeckner)[67]. The wild-type *TLK2* cDNA was introduced into the pcDNA3.1 plasmid generating the TLK2-WT construct. The KD construct TLK2-KD, the phospho-site mutants or mimics S569A, S617A, S617D, S635A, S635D, S686A, S686D, S659A, T695A and S474A and the C-tail mutant S748stop were generated by QuickChange Lightning Site-Directed Mutagenesis (Agilent Technologies). In order to generate domain-deletion mutants, unique EcoRI and ScaI restriction sites were introduced as silent mutations into the TLK2-WT plasmid using the QuickChange Lightning Site-Directed Mutagenesis (Agilent Technologies). ΔN-TLK2 and TLK2-ΔCC3 were generated using these sites to clone in PCR-amplified truncations. TLK2-ΔCC1 and TLK2-ΔCC2 mutants were made by cloning PCR-amplified custom GeneArt DNA-Strings (Thermo Fisher Scientific) into a pCR Blunt End II-TOPO vector (Thermo Fisher Scientific) for storage, amplification and sequencing. The resulting insert was PCR amplified and cloned into the TLK2-WT plasmid to generate TLK2-ΔCC1 and TLK2-ΔCC2 using the EcoRI and ScaI (NEB) sites. Digested insert and plasmid were purified from agarose gels using the PureLink Quick Gel

Extraction and PCR Purification Combo Kit (Thermo Fisher Scientific) and ligated using the Quick Ligation Kit (NEB) in a 3:1 ratio. All constructs were verified by DNA sequencing (primer sequences are shown in Supplementary Tables 4, 5).

**Transfections and pull-downs in mammalian cells**. AD293 cells (Stratagene) were grown in DMEM media (Thermo Fisher Scientific) supplemented with 10% foetal bovine serum (FBS) (Sigma-Aldrich) and 50 U/ml penicillin/streptomycin (Thermo Fisher Scientific) at 37 °C in a 5% $CO_2$ incubator. Transient transfections of 20 μg plasmid per 15 cm plate were carried out using polyethylenimine (PEI) (Polysciences Inc.). Cells were harvested 48 h after transfection and collected by scraping in PBS. Pellets were lysed in 1 ml lysis buffer (50 mM Tris-HCl, pH 7.5, 150 mM NaCl, 1% Tween-20, 0.5% NP-40, 1× protein inhibitor cocktail (Roche) and 1× phosphatase inhibitor cocktails 2&3 (Sigma-Aldrich)) on ice for 20 min. Cells were freeze-thawed three times or sonicated at medium intensity for 10 min, and lysates were cleared by centrifugation at 16,000×g for 20 min at 4 °C. Proteins were quantified using the DC Protein Assay (BioRad) and 100 μl was retained for input; 2–4 mg of total protein extracts were incubated with 100 μl pre-washed Strep-Tactin superflow resin (IBA GmbH) overnight at 4 °C using an overhead tumbler. Resin was spun down (7000×g for 30 s) and transferred to Illustra MicroSpin G-25 Columns (GE Healthcare). Resin was washed three times with 500 μl wash buffer (30 mM Tris, pH 7.4, 150 mM NaCl, 0.1% NP-40, 1× protein inhibitor cocktail (Roche) and 1× phosphatase inhibitor cocktails 2&3 (Sigma-Aldrich)). Proteins were eluted from the Strep-Tactin matrix with 50 μl of 5× Desthiobiotin Elution buffer (IBA GmbH) in TBS buffer (30 mM Tris-HCl pH 7.4, 150 mM NaCl, 1× protein inhibitor cocktail (Roche) and 1× phosphatase inhibitor cocktails 2&3 (Sigma-Aldrich)) for 10 min on ice.

**Western blot and antibodies**. For western blotting, 30 μg of input and 25 μl of Strep-IP elute coming from AD293 cells were separated by SDS-PAGE and transferred to Nitrocellulose membrane (GE Healthcare). Membranes were probed with antibodies for FLAG (Sigma F3165, 1:1000), TLK1 (Cell Signaling 4125, 1:1000), ASF1 (Santa Cruz sc-53171, 1:800 or kindly provided by A. Groth, 1:2000), and LC8 (Abcam ab51603, 1:1000). Primary antibodies were detected with appropriate secondary antibodies conjugated to HRP and visualized by ECL-Plus (GE Healthcare).

**Pull-down kinase assays from cell lysates**. In vitro kinase assays were performed after Strep-IP purification of N-SF-TLK2 mutants expressed in AD293 cells; 200 μg of the Strep-IP eluate was used for the kinase reaction. Reactions were performed in kinase assay buffer (50 mM Tris-HCl pH 7.5, 10 mM $MgCl_2$, 2 mM DTT, 50 mg/ml ATP) and 1× protein inhibitor cocktail (Roche) and 1× phosphatase inhibitor cocktails 2&3 (Sigma-Aldrich), with 1 μg of substrate and $^{32}$P-γ-ATP. After incubating for 30 min at 30 °C, the reaction was stopped by adding 2–3 μl of Sample Buffer (6× SDS, 0.2% bromophenol blue and β-mercaptoethanol) and boiled for 5 min. Samples were analyzed on SDS-PAGE, stained with Coomassie Blue and dried for autoradiography. Substrates used for testing TLK2 mutants were 1 μg of MBP (Sigma-Aldrich) or 0.5 μg of GST-ASF1a (kindly provided by A. Groth). Kinase inhibitors used in Fig. 6 were all obtained from the InhibitorSelect 96-Well Protein Kinase inhibitor library II (Merck).

**Kinase assays with purified proteins**. Standard kinase assay with substrates were performed for 30 min at 30 °C in 10 μl of kinase buffer (10 mM Tris pH 7.5, 50 mM KCl, 10 mM $MgCl_2$, 1 mM DTT) supplemented with 50 μM cold ATP and 1.5μCi [γ-$^{32}$P] ATP (3000 Ci/mmol) with 0–0.5 μM TLK2 proteins and 50 μM of the substrate. Recombinant TLK2 proteins were obtained from over-expressions in *E. coli* as described above. The substrates used for the kinase assays were: bovine dephosphorylated MBP (Merck millipore, Catalogue #13–110), full-length human Histone H3.1 (Sigma-Aldrich, SRP0177 Sigma) and full-length human ASF1a purified as described above. The kinase reaction was stopped by the addition of SDS sample buffer to be further separated by SDS polyacrylamide gel electrophoresis. Polyacrylamide gels were stained with the colloidal staining kit (Invitrogen) and further dried for 1 h 30 min at 80 °C. Radioactive TLK, MBP, Histone H3.1 and ASF1a were identified by autoradiography; quantification of the intensity of the bands on the autoradiograms was achieved by densitometry using the Image Studio Lite Software (LI-COR). Autophosphorylation assays were performed for 1 to 7 min (time course: 10 μl/min) at 30 °C in 80 μl and with 4 μM of the TLK proteins. Sample preparation and quantification of the obtained intensities is the same as for the samples of the substrate kinase assays.

**Immunofluorescence imaging of TLK2 localization**. AD293 cells were transiently transfected with the indicated constructs and fixed for 10 min in 4% paraformaldehyde at room temperature. Following blocking with 3% BSA/0.1% Triton X-100 in PBS for 1 h, cells were stained with a mouse anti-FLAG antibody (1:500, Sigma F1804) and the DNA stain DAPI (Sigma). Images were acquired with oil immersion on a Zeiss LSM 780 confocal microscope at 40× magnification.

**Size-exclusion chromatography–multi-angle light scattering**. The oligomeric state of TLK2 proteins, including the heterodimers, was analyzed by size-exclusion chromatography coupled with multi-angle light scattering (SEC-MALS). Protein samples were prepared at 1 mg/ml concentration and dialyzed into gel filtration running buffer, 20 mM Tris-HCl, 150 mM NaCl, 0.5 mM TCEP, pH 8.0. The samples were loaded on a Superdex 200 Increase 10/300 size exclusion column (GE Healthcare). The column outlet was directly connected to a DAWN HELEOS II MALS detector (Wyatt Technology) followed by an Optilab T-rEX differential refractometer (Wyatt Technology). Data were collected and analyzed using ASTRA 6 software (Wyatt Technology). Samples were run in triplicates. The monomeric BSA was used as standard (Sigma-Aldrich).

**Expression of mammalian TLK2 for MS analysis**. TLK2 full length and ΔN-TLK2 were cloned into the HIS-TwinStrep N-term vector pCPR0197 (LIC-pTT5) using Ligation Independent Cloning (LIC). The constructs were verified by colony PCR followed by sequencing. The plasmids were transformed into Mach1-T1 cells (Invitrogen) and 2.7 litres overnight cultures were GIGAprepped with the Nucleobond PC 10000 EF kit (Macherey-Nagel). The constructs were boiled at 95 °C for 5 min before transfection. The transfection was performed into Human Embryonic Kidney EBNA 6E cell lines (HEK293 6E) grown in Freestyle 293 F17 expression medium (Invitrogen) + 4 mM L-glu (Sigma). One day prior to transfection, HEK293 6E cells were re-suspended in fresh Freestyle 293 F17 expression medium + 4 mM L-glu to a cell density of $1.2 \times 10^6$ cells/ml and incubated at 37 °C overnight. Approximately, 15 min before transfection, cells were re-suspended in fresh unsupplemented Freestyle 293 F17 expression medium + 4 mM L-glu at a cell density of $20 \times 10^6$ cells/ml and incubated in the orbital shaker incubator at 37 °C, 70% humidity, 5% $CO_2$ and 120 RPM (Ø50 mm), until being transfected. GIGA-prep DNA of each construct (50 μg/ml final) and Polyethylenimine "MAX" (PEI) (polysciences) (100 μg/ml solution final) were directly added to the cell suspension. Complete Freestyle 293 F17 expression medium (1% FBS) was added to a final volume of 2 litres of cell suspension, 4 h post transfection. Three days post transfection, the pellets were collected by centrifugation at 4000 RPM for 15 min at 4 °C. The pellets were washed with PBS at 4 °C and stored at −20 °C. TLK2 proteins were purified using Strep-tag/Strep-Tactin purification system provided by IBA. Prior to the purification, the pellet was re-suspended in the appropriate lysis buffer. The clear lysate was centrifuged at 15,000 RPM for 30 min at 4 °C and filtered (Minisart 0.22 μm filter). The fractions were loaded on a NuPAGE Bis-Tris gel to check for protein integrity and purity.

**Sample preparation for MS analysis**. For MS-based analysis of TLK2 phosphorylation status, the protein was resolved by sodium dodecyl sulphate-polyacrylamide gel electrophoresis (SDS-PAGE), visualized by Coomassie staining and in-gel trypsin digested (protocol adapted from Lundby and Olsen)[68]. Prior to SDS-PAGE, proteins were reduced with 10 mM dithiothreitol (DTT) in 25 mM ammonium bicarbonate (ABC) buffer for 45 min and alkylated with 55 mM chloroacetamide (CAA) in 25 mM ABC solution for 30 min. SDS-PAGE was performed with the different TLK2 constructs using 4–12% bis-tris gradient gels (Invitrogen) and stained with the Colloidal Blue Kit (Invitrogen) according to manufacturer instructions. For each sample, the TLK2-specific band was excised from the gel in 1 × 2-mm cubes. Gel slices were destained with 50% ethanol in 25 mM ABC solution and dehydrated with 96% ethanol. Proteins were digested with trypsin (modified sequencing grade, Sigma) overnight at 37 °C. Trypsin activity was quenched by acidification with trifluoroacetic acid (TFA), and peptides were extracted from the gel pieces using increasing concentrations of acetonitrile. Organic solvents were removed by evaporation in a SpeedVac centrifuge at 60 °C and samples were desalted and concentrated by solid-phase extraction on reversed-phased $C_{18}$ STAGE tips.

**Liquid chromatography–tandem MS**. For all samples, peptides were eluted from the $C_{18}$ STAGE tips with 40% and 60% acetonitrile in 0.1% formic acid prior to online nanoflow LC-MS/MS analysis. Samples were analyzed with a nanoscale UHPLC system (EASY nLC1200, Thermo Fischer Scientific) connected to a Q Exactive HF mass spectrometer (Thermo Fischer Scientific) through a nanoelectrospray ion source as previously described[69]. Briefly, peptides were separated in a 15-cm analytical column (75-μm inner diameter) in-house packed with 1.9-μm reversed-phase $C_{18}$ beads (ReproSil-Pur AQ, Dr. Maisch) with a 76 min gradient from 8 to 64% acetonitrile in 0.5% formic acid with a flow of 250 nl/min. The mass spectrometer was operated with spray voltage set to 2 kV, heated capillary temperature at 275 °C and s-lens radio frequency level at 50%. Dynamic exclusion was set to 30 s and all experiments acquired in positive polarity mode. Full scan resolution was set to 120,000 at *m/z* 200, and the mass range was set to *m/z* 375–1500. Full scan ion target value was 3E6 with a maximum fill time of 25 ms. For every full scan, the 12 most intense ions were isolated and fragmented (normalized collision energy 28%) by higher-energy collisional dissociation (HCD) with fragment scan resolution of 30,000, and an ion target value of 1E5 with a maximum fill time of 45 ms.

**Processing and analysis of MS raw data.** All raw LC-MS/MS data files were analyzed using the MaxQuant software version 1.5.3.36 with the integrated Andromeda search engine[70,71]. Data were searched against a target/decoy (forward and reversed) version of the complete human UniProt database supplemented with commonly observed contaminants and the sequences of the TLK2 constructs. In addition, the ECOLI database was employed in the search to identify potential co-purifying host cell proteins. Cysteine carbamidomethylation was searched as a fixed modification. Protein N-terminal acetylation, oxidized methionine, pyroglutamate formation from glutamine, and phosphorylation of serine, threonine and tyrosine were searched as variable modifications. In addition, deamidation of asparagine and glutamine was searched as an extra variable modification. Phosphorylation site localization probabilities and occupancies[72] were determined by MaxQuant using the PTM scoring algorithm[70]. An initial false discovery rate (FDR) of 1% and 5%, respectively, was applied for peptide and phosphorylation site identifications.

**Bioinformatic data analysis.** Analysis of proteomics data was performed using the Perseus software version 1.5.1.12 (http://www.coxdocs.org/). Only peptides with a phosphorylation site localization probability of at least 0.75 (class I sites)[73] were included in the final bioinformatics analyses. Phosphorylation site identifications were filtered for contaminants and reversed hits. Data were filtered based on the criteria that a phosphorylation site had to be identified in at least two replicates for at least one of the sample groups in order to be included in the downstream analysis. Data analysis was done using the raw intensity values, which were log2-transformed and normalized by quantile-based normalization and median subtraction. Imputation from the lower end of the normal distribution was done to replace missing values. A heat map was generated based on unsupervised hierarchical clustering analysis using the Perseus software. Phosphopeptides and phosphorylation sites significantly regulated between the different TLK2 constructs were determined using Student's t-test by comparing intensities for the specific constructs and results of the analysis were visualized by volcano plots.

**HADDOCK docking targeting the ATP-binding pocket.** HADDOCK version 2.2[54] using CNS[74] for structure calculations was used to dock the three selected ligands (CGP74541A, GSK3 inhibitor XIII and the indirubin derivative E804) in the ATP-binding pocket of the TLK2 kinase domain. The crystal structure of the TLK2 kinase domain in complex with ATPγS was used to extract the initial coordinates of the protein for the docking and to identify the residues defining the ATP-binding pocket. Kinase domain residues with atoms with centres within 10 Å of the centre of any atom of ATPγS and with all-atom relative solvent accessibility greater than 20% define the ATP-binding pocket. This comprises the protein residues H466-K478 (from β1, β2 strands and the glycine-rich P-loop), Y487-Q494 (β3 strand), Y507, H510, A511 and E514 (αC helix), V527-Y530 (β4 strand), L544-D551, D553 and F554 (β5 strand, αD helix), D592, K594-L599 and V601 (catalytic loop), K610, T612, D613, G615 and L616 (magnesium binding loop). The ligand coordinates for the docking were extracted from the crystal structures of the following complexes: human P21-activated kinase 4 in complex with CGP74541A (PDB entry 2cdz), human calmodulin-dependent protein kinase 1D in complex with GSK3 inhibitor XIII (PDB entry 2jc6) and human calcium calmodulin-dependent protein kinase type II alpha (CAMK2A) in complex with indirubin E804 (PDB entry 2vz6). The default version topology and parameter files provided for proteins in HADDOCK 2.2 were used for generation of the protein and ligand structures. Histidine residues were protonated per the default setting in HADDOCK; 1000 docked complex structures were generated in the first rigid-body docking step (it0), 200 structures in the semi-flexible simulated annealing (it1) and 200 structures evaluated in the final analysis. Two set of distance restraints were used at different stages of the docking protocol. For the rigid body docking, it0, the entire binding pocket and the ligand were defined as active. For the flexible refinement steps, the binding pocket was defined as passive while the ligand was defined as active. Molecular dynamics simulations were switched off (the number of MD steps was set to 0) for both rigid-body high-temperature docking and the slow cooling annealing step of the semi-flexible simulated annealing. HADDOCK score was used to rank the models. Models were clustered using the RMSD criteria with a 2.0 Å cut-off. The structures of the top clusters with lowest HADDOCK scores were examined manually in Pymol.

**Data availability.** Structure factors and coordinates have been deposited at the PDB (PDB code 5O0Y). The MS raw data and associated tables have been deposited in the ProteomeXchange Consortium via the PRIDE partner repository with the dataset identifiers PXD007675 and PXD009095. Other data are available from the corresponding author upon reasonable request.

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

## Acknowledgements

Novo Nordisk Foundation Center for Protein Research is supported financially by the Novo Nordisk Foundation (Grant NNF14CC0001) and the Danish Cancer Foundation for a grant to G.M. T.H.S. is supported by the Ministerio de Economía y Competitividad (MINECO) (BFU2015-68354, Ayudas para incentivar la incorporación estable de doctores (IED) 2015 and institutional funding through the Centres of Excellence Severo Ochoa award and from the CERCA Programme of the Catalan Government), S.S.B. by a predoctoral fellowship from Fundacio La Caixa and M.V.P. by a Severo Ochoa FPI predoctoral fellowship (MINECO). P.R. is supported by the Marie Skłodowska-Curie European Training Network (ETN) "TEMPERA. We would like to thank the Protein Production Facility Platform at CPR for the excellent technical assistance, M. Orozco at IRB Barcelona for helping with the initial modelling analysis and also thank the PRO-MS Danish National Mass Spectrometry Platform for Functional Proteomics and the CPR Mass Spectrometry Platform for instrument support and assistance.

## Author contributions

A.M.G. and P.R. expressed and purified the kinase domain constructs. G.B.M. expressed and purified the rest of the constructs and performed the characterization with I.P. and D.H. and coordinated that part of the project. D.H. performed all the activity assays with the isolated proteins from bacteria and the mathematical analysis together with I.P. A.M.G., I.G.M. and G.M. solved the structure. G.M. performed the structure refinement and model building and G.M.B. deposited the coordinates. A.K.P., Pa.R. and J.V.O. performed the mass spectrometry experiments, statistical analysis and interpreted the data together with G.B.M. and G.M. S.S.B. and C.J. carried out the mutagenesis of the kinase domain and analyzed expression and activity from human cell lysates. M.V.P. performed the IF analysis of mutant TLK2 expression and localization in human cells. B.L.M. performed the modelling and its analysis with G.M. G.M.B. prepared the data and wrote the draft of the manuscript. T.H.S. wrote and edited portions of the manuscript, analyzed the data and designed and supervised the experiments. G.M. conceived the project with input from all the authors and supervised the experiments, besides coordinating the writing of the manuscript.

## Additional information

**Competing interests:** The authors declare no competing interests.

