## [Peer Review File · Nature Communications]

Reviewers' comments:

Reviewer #1 (Remarks to the Author):

The manuscript by Mortuza et al "Molecular basis of Tausled-Like Kinase 2 activation" reports the structure of TLK2 kinase at about 2.9 Å resolution. Analysis of recombinant protein constructs revealed that TLK2 containing the C-terminal CC domain, but not the kinase domain alone can dimerize. CC-domain dependent dimerization enhances activity of the recombinant protein probably by enhanced autophosphorylation which also promotes assembly of higher order oligomers.

Recombinant TLK2 is active and phosphorylates the general substrate MBP as well as specific substrate ASF1a. The latter phosphorylation requires the CC domains. The authors also mapped a large number of auto-phosphorylation sites of TLK2 expressed in the presence or absence of lambda phosphatase.

The structure of the kinase domain reveals a typical kinase domain fold with large deviations to observed structures observed in the activation loop and other mobile elements. Finally, the authors generated disease related mutants observed in intellectual disability identifying a number of loss of function mutations in particular D629A. Furthermore, known TLK2 kinase inhibitors were docked into the ATP binding site to generate a model for future drug design efforts.

This represents the first structure of TLK2, however, the structure covers only the catalytic domain a medium resolution limiting the usefulness of the structural model for functional understanding of this kinase as well as the use of the structure as a template for inhibitor design. The dimerization domain for instance is not included in the model making it difficult to understand the molecular mechanisms of activation. Furthermore, phosphorylation sites have been mapped on recombinant protein only and the role of the mapped sites and if they occur in vivo is not clear. I suggest therefore publication of the data in a more specialized journal.

Minor issues: The diffraction data should be limited to a reasonable resolution. In the last shell (2.96-2.86 Å) the dataset has a completeness of 1 % with a mean $I/\sigma I$ of 0.54. I suggest cutting the data at a mean $I/\sigma I$ of about 2.0. The Average B-factor is > 100 (for the ligand it is 125). Thus, it is not likely that the ligand could be confidently modelled into the density. I suggest showing a figure with a 3FoFc omitted map to convince the reader of the binding mode of the ligand.

Reviewer #2 (Remarks to the Author):

The manuscript by Mortuza et al describes a detailed molecular mechanism of TSLK2 activation. The authors have employed a multitude of different techniques to assess this mechanism and provide convincing data.

The manuscript is well written and the figures are well designed. I think the paper represents a good example of how to solve complex signaling mechanisms using a variety of techniques. There are some issues that need to be addressed however.

For the heterodimer analysis, this experiment seems to be a bit complicated. Although the authors can separate the KD from the WT using different tags, in the experiment itself they cannot exclude homodimer formation. If that is correct, only half of WT and KD would actually engage in a heterodimer (and 1/4 WT-WT, 1/4 KD-KD). After SDS page the WT and KD will be separated due to different tags, however the signal of the phosphopeptide will be the average of the different dimers formed. In that context some of the detected sites could still be trans (for WT homodimer) and more importantly the reported ratios are not an accurate reflection of the heterodimer state, but contain homodimer signal. The authors should address this. The same actually goes for Figure 2C.

Minor issues

Figure 3a, this heatmap does not have a legend. It would be good to add a legend that indicates the fold change.

For the MS data it would be good if the authors would state somewhere what the confidence of the identified sites is. Most of the putative sites are located quite closely to each other which would make localization extremely difficult. Some assessment of the confidence would be good. Especially for follow up work this is extremely important.

Figure 3, last line of the legend, volcano plots are in Supp Fig 5 (not 6 as mentioned here)

The table is not correctly converted to pdf format and cannot be checked.

Reviewer #3 (Remarks to the Author):

Molecular basis of Tausled-Like Kinase 2 activation

The authors report the crystal structure of the catalytic domain of TLK2. What makes the study important is their analysis of TLK2 phosphorylation on activation which involves a combination of oligomerization and auto-phosphorylation in cis and trans. A preliminary report on the crystallization appeared in Crystallization Communications in 2014 by some of the authors.

Specific points

1 Page 10 line 241. A number of phosphorylation sites on TLK2 are on Phosphosite. In particular the authors report multiple phosphorylation sites in the loops around the CC domains between residues 190-450. Since these have not appeared in the Phosphosite an important question is the stoichiometry of these sites and whether they occur naturally or are confined to E coli expressed truncated enzyme. It would be important to show these sites are present or not present in mammalian cultured cells at least. Can the authors provide TOF data to provide a measure of the stoichiometry and maximum numbers of sites.

2 Page 12 line 287. Is there time-course information to show which sites are phosphorylated first and how the activation mechanism operates in terms of the sequence of events?

3 Page 13 line 306. The authors refer to non-hydrolysable ATP analogue ATP γ S. If the sulphur is on the γ -phosphate rather than between the β and γ phosphates it should be hydrolysable as ATP γ S has been used for decades to thio-phosphorylate proteins to make them resistant to phosphatases. Line 327 in reference to the HRD do the authors mean TLK2 has YDL instead. CaMK-II has an HRD and it does not require phosphorylation.

4 Page 16 line 386. The naturally occurring mutation is D629N and the authors have tested D629A. It would be important to test the correct mutation.

5 Page 17 line 413. The authors report docking of a few kinase inhibitors as an example of the potential utility of their structure in developing inhibitors for TLK2. The modelling is not covered in the methods. The authors have used a strategy of choosing solutions without clashes. However, kinases are flexible. Did the authors try minimization of the solutions and or test alternate conformations? It would have been useful to have done the modelling with more than one software package to get a consensus result. Overall the modelling provides a rather weak end to the manuscript and could be deleted as it detracts from the novelty of the mutation analysis.

6 Page 26-27. Methods or an appropriate reference for growing TLK2 crystals was not present in this section.

7 Page 40 - Table 1 statistics indicate resolution range has been greatly exaggerated. The authors

need to re-assess the resolution cut-off, likely to be closer to 3 Å or more. Most worrying aspects are the completeness in the highest resolution shell (HRS) is only 1.0 %, Rmerge and Rmeas (HRS) are very high (typically below 0.5), Mean I/sig(I) and CC1/2 (HRS) are below reasonably accepted standards of 2.0 and 0.5 respectively.

Also missing a bracket in the total reflections row.

Point-by-point response to review comments:

Reviewer #1

We thank the reviewer for the comments and the corrections/suggestions to improve our manuscript. We have corrected all of the minor points found by the referee and we have introduced many of the suggestions. Nevertheless, we disagree with his/her main criticism regarding the resolution of our structure. We have addressed this and the other comments in our rebuttal.

This represents the first structure of TLK2, however, the structure covers only the catalytic domain a medium resolution limiting the usefulness of the structural model for functional understanding of this kinase as well as the use of the structure as a template for inhibitor design.

This is the first structural information of the TLK2 family kinase domain, as the reviewer points out. The resolution of the structure is 2.86 Å, which is enough to visualise all the side chains, as well as the ATP γ S molecule used in co-crystallisation. There are more than 100,000 models at this resolution currently in the PDB database (see PDB report of our structure). Many of them are of kinases that have been used in inhibitor design. For example, the Aurora A kinase structure has been used for inhibitor design for its targeting in cancer. All of the following PDBs are structures of AuroraA with ligands between 2.8 and 3.5 Å resolution: 2x6d, 2x6e, 2x81 3h0z, 3unz, 3uo6, 3uoj, 3uoh...etc. and many more for AuroraA. This is also true for other kinases that I do not include here for the sake of brevity. Therefore, we strongly believe that our structure is useful for understanding the biological regulation of the TLKs and for use in inhibitor design, as other kinase domain structures at similar resolution have clearly proven useful for this purpose. In fact, this is supported by the fact that we have been able to model some of the current known inhibitors using our structure as shown in Fig. 6.

In addition, the PDB report, that is available to the reviewer, shows clearly that the statistical parameters indicate that 91% of the residues display a good fit to the electron density (see plots on Pag.2 and 4 of the PDB report). Only a few residues in the N- and C-tail, where the chain is more flexible, give some trouble. We think that the statistics and the PDB report are enough to demonstrate the quality of the data and therefore these criticisms cannot be sustained. Nevertheless, to further support our data, and to clarify any possible confusion, we have included in the revised version a 2Fo-Fc electron density map (Supp. Fig 7) where the reader will be able to assess the quality of the electron density map. We therefore feel that these figures that include the density maps address this point in full.

The dimerization domain for instance is not included in the model making it difficult to understand the molecular mechanisms of activation.

Although we have not been able to crystallise the full-length protein, we provide the first crystal structure of the TLK kinase family, helping to complete the structural landscape of this branch of the human kinome. Indeed, the structure shows an atypical T-loop structure (see new Supp. Fig 7), which is unique to this family and, as mentioned before, this is important information regarding the TLK family. Moreover, the future analysis of the structure and regulation of the full-length TLK family will also benefit from the biochemical analysis performed in our manuscript to help fully unravel the activation mechanism.

Furthermore, phosphorylation sites have been mapped on recombinant protein only and the role of the mapped sites and if they occur in vivo is not clear. I suggest therefore publication of the data in a more specialized journal.

To understand the mechanism of TLK2 activation, we sought to identify the phosphorylation sites of TLK2 in a controlled system. Thus, we have performed the analysis expressing the protein in *E. coli*. Experiments in the revised version of the manuscript describe the

phosphosite mapping of TLK2 expressed in mammalian cells (HEK293), and show that TLK2 purified from these cells shows a similar phosphorylation pattern as TLK2 purified in the *E.coli* system. This clearly indicates that many of the sites we previously observed occur in mammalian cells

Our experimental approach has allowed us for the first time to shed light on how the TLK2 dimer is activated. Whether every site observed *in vitro* plays a key role *in vivo* is not experimentally feasible to address, even in the mammalian expressed protein. It is worth mentioning that only 41 out of 67 possible phosphorylation sites have been detected. Therefore, around half of the Ser/Thr residues of TLK2 have not been modified. Although we cannot attribute a role for all, our approach shows clearly that several of the residues involved influence kinase activity and substrate modification in TLK2 expressed in human cells (Fig. 5b and Supp. Fig. 8a-b). Even in the case that we would have had the complete structure of the dimerization domains, we would need these mass spectrometry experiments in Fig. 3 to fully understand the activation mechanisms.

Minor issues: The diffraction data should be limited to a reasonable resolution. In the last shell (2.96-2.86 Å) the dataset has a completeness of 1 % with a mean I/sigma I of 0.54. I suggest cutting the data at a mean I/sigma I of about 2.0.

This minor point arises from confusion due to a typo in the crystallographic table, turning the 100% in that shell to 1.00%. We apologize for the confusion. Given that the completeness of the last resolution shell is 100% and using the CC(1/2) criterion, we do not agree with the referee in the suggested resolution cut. It has been demonstrated that the new processing programs together with the new pixel array detectors can make use of weak diffraction data avoiding the conservative 2.0 I/sigmaI cut-off applied some years ago, please see the following recent papers, where the theory behind this new data handling and mathematic treatment is discussed in detail for reference:

Karplus PA, Diederichs K. Linking crystallographic model and data quality. Science. 2012 May 25;336(6084):1030-3.

Diederichs K, Karplus PA. Better models by discarding data? Acta Crystallogr D Biol Crystallogr. 2013 Jul;69(Pt 7):1215-22.

Karplus PA, Diederichs K. Assessing and maximizing data quality in macromolecular crystallography. Curr Opin Struct Biol. 2015 Oct;34:60-8.

Therefore, our data are up to the current quality standards and we think that the structure is extremely useful, with a resolution reliable for its use in guiding biochemical analysis and drug design.

The Average B-factor is > 100 (for the ligand it is 125). Thus, it is not likely that the ligand could be confidently modelled into the density. I suggest showing a figure with a 3FoFc omitted map to convince the reader of the binding mode of the ligand.

The ligand can be perfectly modelled in the electron density, as observed in the requested map that is shown in the figure below contoured at 1.2 sigma. We have included an omit map in the Fig 4c of the revised version of the manuscript and it can also be observed in the new Supp. Fig. 7.

Reviewer #2

We thank the reviewer very much for the positive comments and the corrections/suggestions to improve our manuscript. We have corrected all the minor points found by the referee and we have introduced many of the suggestions. The main query regarding the heterodimer experiment can be easily explained and we will introduce modifications in the text to clarify that point.

For the heterodimer analysis, this experiment seems to be a bit complicated. Although the authors can separate the KD from the WT using different tags, in the experiment itself they cannot exclude homodimer formation. If that is correct, only half of WT and KD would actually engage in a heterodimer (and 1/4 WT-WT, 1/4 KD-KD). After SDS page the WT and KD will be separated due to different tags, however the signal of the phosphopeptide will be the average of the different dimers formed. In that context, some of the detected sites could still be trans (for WT homodimer) and more importantly the reported ratios are not an accurate reflection of the heterodimer state, but contain homodimer signal. The authors should address this. The same actually goes for Figure 2C.

Perhaps we have not stated it clearly enough in the manuscript but we have not detected interconversion in the scale of time used for the experiments. We checked this point and the heterodimer is stable for days). Therefore, it is not the case that once the heterodimer is isolated there is dissociation and interconversion between the monomers. The monomeric form of the full-length construct does not seem to be stable. In addition, this is supported by our mass spec measurement in Fig. 3d, where we observed a differential phosphorylation pattern between the WT and KD monomers. If interconversion would happen after isolation we would have not been able to detect an asymmetric phosphorylation arrangement. Consequently, this will not affect the phosphopeptide detection in Fig. 3d or the experiment in Fig. 2c. We have introduced modifications in the text to clarify this point (In the Results section for Fig 1 and 3) and avoid confusion.

Minor issues

Figure 3a, this heat map does not have a legend. It would be good to add a legend that indicates the fold change.

We have corrected this point

For the MS data it would be good if the authors would state somewhere what the confidence of the identified sites is. Most of the putative sites are located quite closely to each other which would make localization extremely difficult. Some assessment of the confidence would be good. Especially for follow up work this is extremely important.

Regarding this point, this is already stated in the materials and methods section under bioinformatics analysis. We only included sites with a phosphorylation site localisation probability of at least 0.75 (class I sites; explained in reference 71 in the paper) in the final bioinformatics analyses. This means that for sites reported in the final list, the phospho-group is localized to the specific site with a localization probability score of minimum 0.75. The specific scores for each site are in the Excel tables provided, that unfortunately were not properly converted for the initial review. We apologize for this issue and have included the properly formatted excel file to avoid confusion. Let us know through the editor if there is any problem with the format again.

Figure 3, last line of the legend, volcano plots are in Supp Fig 5 (not 6 as mentioned here)

We have corrected this point

The table is not correctly converted to pdf format and cannot be checked.

The table was deposited as an excel file and did not convert correctly. We have specified this point in the new submission, please let us know through the editor if you have any problems accessing this in the proper format.

Reviewer #3

We thank the referee very much for the positive comments and the corrections/suggestions to improve our manuscript. We have addressed the points raised by the reviewer and corrected all the minor points found by the referee.

Specific points

1 Page 10 line 241. A number of phosphorylation sites on TLK2 are on Phosphosite. In particular the authors report multiple phosphorylation sites in the loops around the CC domains between residues 190-450. Since these have not appeared in the Phosphosite an important question is the stoichiometry of these sites and whether they occur naturally or are confined to E coli expressed truncated enzyme. It would be important to show these sites are present or not present in mammalian cultured cells at least. Can the authors provide TOF data to provide a measure of the stoichiometry and maximum numbers of sites.

We have performed our analysis in the *E.coli* expressed sample because this is a very controlled system, as we have shown in our manuscript, free of phosphatases and other kinases that could contaminate our analysis. However, the high expression in the prokaryotic system could lead to the phosphorylation of secondary sites that may have low or no functional role. Therefore, the referee has raised a fair question because many of the phosphosites in the 191-450 region have not been previously observed. To address this question, we have expressed the (191-772) and full length TLK2 constructs in HEK293 cells. As can be observed in the new Fig 3d, new supp fig 5a-b and the Excel file, out of the 25 sites detected in this region in the *E.coli* expressed TLK2, 15 can be also found in the HEK293 expressed sample. Therefore, a large number of the new phosphosites sites reported in the loops around the CC domains can be also found in the HEK293 expressed protein, especially in the 202-330 segment. Therefore, these regions are also found phosphorylated in mammalian cells. The experiment has been repeated 3 times and the level of detection is well above a stringent statistical cut-off.

In the table “PR180220_TLK2_PhosphoSites”, there are stoichiometries on sheet 4. They are called “median occupancies”, which is the percentage of stoichiometries comparing the phosphorylated form with the non-phosphorylated counterpart. For instance, an occupancy of 50 % means a 1:1 stoichiometry. We have a median occupancy of ~50 % in the case of the wildtype full-length and truncated form and a median occupancy of only 15 % in the case of two the kinase-dead versions.

This result reinforces our analysis, and we have modified the text and the figure accordingly to introduce these new data in the revised version of the manuscript, also pointing out the minor differences in the phosphorylation pattern between the (191-772) and full length TLK2 constructs in HEK293 cells.

2 Page 12 line 287. Is there time-course information to show which sites are phosphorylated first and how the activation mechanism operates in terms of the sequence of events?

No, we do not have time-course information. Therefore, we have rephrased that section to give that impression and avoid confusion.

3 Page 13 line 306. The authors refer to non-hydrolysable ATP analogue ATPγS. If the sulphur is on the γ-phosphate rather than between the β and γ phosphates it should be hydrolysable as ATPγS has been used for decades to thio-phosphorylate proteins to make them resistant to phosphatases.

The reviewer is correct, The ATP analogue adenosine 5'-(gamma-thiotriphosphate) can be hydrolysed, although rather slowly. Therefore, we have modified that sentence accordingly.

Line 327 in reference to the HRD do the authors mean TLK2 has YDL instead. CaMK-II has an HRD and it does not require phosphorylation.

The referee is shifted by one residue in the sequence (see supp. fig 1 alignment), TLK2 displays HYD and no HRD. Furthermore, we have not found phosphorylation of the S635. Most RD-kinases require a phosphorylation on their activation loop for its proper activation. However, the reviewer is right in pointing out that CAMK-II does not require activation loop phosphorylation, it is one of the exceptions to this observation. We have rephrased this section in the new version

4 Page 16 line 386. The naturally occurring mutation is D629N and the authors have tested D629A. It would be important to test the correct mutation.

Following the referee comments, we have tested the correct mutation, D629N, and modified the results and discussion according to the new results.

5 Page 17 line 413. The authors report docking of a few kinase inhibitors as an example of the potential utility of their structure in developing inhibitors for TLK2. The modelling is not covered in the methods. The authors have used a strategy of choosing solutions without clashes. However, kinases are flexible. Did the authors try minimization of the solutions and or test alternate conformations? It would have been useful to have done the modelling with more than one software package to get a consensus result. Overall the modelling provides a rather weak end to the manuscript and could be deleted as it detracts from the novelty of the mutation analysis.

We apologise for not including the materials for the modelling section. A thorough description of the modelling is included in materials and methods in the revised version of the manuscript. We think that a deep modelling analysis could be the aim of an additional manuscript. It was not our intention to make it in this manuscript. Taking advantage of our structure we wanted to explain how the molecules used experimentally to inhibit TLK2 may function. This purpose is well supported by our data in this manuscript. We will rephrase some of the paragraphs to avoid overstatements and include a sentence saying that a more extensive analysis is necessary to fully understand the binding of small molecules to the TLK2 domain.

6 Page 26-27. Methods or an appropriate reference for growing TLK2 crystals was not present in this section.

All these points are described in detail in reference (46) Garrote et al., Therefore, we think that there is no need to include them here.

7 Page 40 - Table 1 statistics indicate resolution range has been greatly exaggerated. The authors need to re-assess the resolution cut-off, likely to be closer to 3 Å or more. Most worrying aspects are the completeness in the highest resolution shell (HRS) is only 1.0 %, Rmerge and Rmeas (HRS) are very high (typically below 0.5), Mean I/sig(I) and CC1/2 (HRS) are below reasonably accepted standards of 2.0 and 0.5 respectively. Also missing a bracket in the total reflections row.

As we previously pointed out in the response to Reviewer 1, this minor point arises from confusion due to a typo in the crystallographic table, turning the 100% in that shell to 1.00%. We apologize for the confusion. Given that the completeness of the last resolution shell is 100% and using the CC(1/2) criterion, we do not agree with the referee in the suggested resolution cut. As previously mentioned, it has been demonstrated that the new processing programs and the new pixel array detectors can make use of weak diffraction data avoiding the conservative 2.0 I/sigmaI cut-off, please see the following recent papers, where the theory behind this new data handling and mathematical treatment is discussed in detail.

Karplus PA, Diederichs K. Linking crystallographic model and data quality. Science. 2012 May 25;336(6084):1030-3.

Diederichs K, Karplus PA. Better models by discarding data? Acta Crystallogr D Biol Crystallogr. 2013 Jul;69(Pt 7):1215-22.

Karplus PA, Diederichs K. Assessing and maximizing data quality in macromolecular crystallography. Curr Opin Struct Biol. 2015 Oct;34:60-8.

Therefore, we believe that our diffraction data clearly meet the current quality standards. The

quality of the maps can be observed in the omit map in fig 4 and the supp. Fig. 7 in the revised version of our manuscript.

Reviewers' comments:

Reviewer #2 (Remarks to the Author):

The authors have adequately responded to all questions raised. Especially the additional phosphorylation analysis in HEK293 cells has made the manuscript stronger.

Reviewer #3 (Remarks to the Author):

Page 5 Results line 6 the authors should specify the phosphorylation site residue not "one". How is reference 18 relevant for citing here?

Page 11 second para line 5. It is not clear how the authors know whether heterodimer interconversion occurs after purification.

Fig 1b Why does the left y-axis have MW units?

Fig 3a Looking across the data there is quite a bit of variability. For example, in the Delta-N-TLK2 samples 29 out of the 40 or so p-sites show differences between the triplicates.

Response to Item 1 The phosphorylation data remains indigestible and difficult to interpret. Fig 3 (a) is not listed in the legend. Panel b appears to consist of only black and red dots for phosphorylation sites, where are the orange labelled sites? In panel c it says boxed sites are ones found in both E. coli and HEK293 cells, but it is not clear what is the origin of the unboxed sites are. In the legend it is not clear what the green versus black versus red labelling of the sites represents. The Excel file provided does not have a legend supplied. Occupancy values are given but how are these determined. Frequently the pattern of tryptic cleavage of a dephosphorylated site is different from the cleavage around a phosphorylation site. Although SD values are listed the significance has not been calculated. While an average occupancy of 47%, the average SD is 12%. The point of requesting a TOF profile of the TLK2 was to see if the dominant species had on average a few phosphates or 10 or more. This would be a trivial experiment for the authors to undertake.

One of the concerns about the numerous phosphorylation sites reported here is the possibility that many are artefacts of over expression in E coli or HEK293 cells. If one inspects the p-sites recorded in Phosphosite the most prominent (number of reports from HTP studies) occur in the first 1-185 residues with few in the region 191-771. There are only 8 sites (S223, Y259, S752, S753, S763, S769, S770 and S771) recorded in Phosphosite compared with ~40 sites reported here. The occupancy of the shared sites between Phosphosite and the authors are S223 (67%), S770 (81%) and S771 (45%).

In order to gain insight into the significance of the phosphorylation sites identified it would be important to know whether they occur under native/in vivo conditions. Second it would be important to know if at least one or preferably a few p-sites had functional roles in vivo. Without this information the authors are not providing a sufficient advance on the importance of dimerisation and the mechanism of activation.

Reference 60 needs updating to Leroux AE, Schulze JO, Biondi RM. AGC kinases, mechanisms of regulation and innovative drug development. *Semin Cancer Biol.* 2018 Feb;48:1-17. doi: 10.1016/j.semcancer.2017.05.011.

Reviewer #3

We thank the referee very much for the positive comments and the corrections/suggestions to improve our manuscript. We have addressed the points raised by the reviewer and corrected all the minor points found by the referee.

Page 5 Results line 6 the authors should specify the phosphorylation site residue not "one". How is reference 18 relevant for citing here?

We have corrected the text to indicate that the site is S743 in TLK1. We have removed reference 18, the reviewer is correct that this is not pertinent to the regulation of TLK2 by phosphorylation and was added in error.

Page 11 second para line 5. It is not clear how the authors know whether heterodimer interconversion occurs after purification.

This is mentioned in Pag. 6 and now also in material and methods in the size-exclusion chromatography–multi-angle light scattering section, we have stated now that the heterodimers were also checked using this method.

Fig 1b Why does the left y-axis have MW units?

We have removed the MW and show molar masses (g/mol) for corresponding chromatographic peaks and the elution profiles. They are shown in the sample plot, see the lines associated to the peaks and the Wyatt website if further explanations are needed. <https://www.wyatt.com/solutions/techniques/sec-mals-molar-mass-size-multi-angle-light-scattering.html>

Fig 3a Looking across the data there is quite a bit of variability. For example, in the Delta-N-TLK2 samples 29 out of the 40 or so p-sites show differences between the triplicates.

When replacing “NA” values with background noise from the left end of the normal distribution, these will be lower than most measured intensities and therefore make phosphosites with missing values appear more variable. Imputation of missing values is a standard procedure in proteomics and was performed using the Perseus software as specified in the material and methods. This is also explained in the web site which is included in Materials and Methods <http://www.coxdocs.org/>. Also, in the reference below.

Tyanova, Stefka, Tikira Temu, Pavel Sinitcyn, Arthur Carlson, Marco Y. Hein, Tamar Geiger, Matthias Mann, and Jürgen Cox. 2016. “The Perseus Computational Platform for Comprehensive Analysis of (prote)omics Data.” Nature Methods 13 (9): 731–40.

To assess the variability, we calculated the Pearson correlation for the phosphopeptide intensities in each set of triplicates. We obtained an average Pearson correlation between replicates of 0.90 for the E.coli dataset and 0.75 for the HEK293 dataset. We factor in this degree of variability by doing statistical testing (t-tests, volcano plot).

Response to Item 1 The phosphorylation data remains indigestible and difficult to interpret. Panel b appears to consist of only black and red dots for phosphorylation sites, where are the orange labelled sites? In panel c it says boxed sites are ones found in both E. coli and HEK293 cells, but it is not clear what is the origin of the unboxed sites are.

We apologize for not making this clearer in the previous version. We have rewritten this fig. legend to make it easier for the reader to understand. In addition, we have changed the colour of the orange dots to light orange and we have changed accordingly the colour of the fonts in 3b to follow the same scheme. We have also modified the figure to make the origin of the different sites in 3b and their relation with 3c easier to understand.

Fig 3 (a) is not listed in the legend. In the legend it is not clear what the green versus black versus red labelling of the sites represents.

There must be some confusion. When we download the files from the website, Panel 3a is in the Fig. 3 legend (page 40 in the PDF and word versions of the document uploaded in the Nat. Comms. Website)

The figure title is “Hierarchical clustering analysis of the phosphorylation sites of the Δ N-TLK2 constructs expressed in E.coli displayed in a heat map”. There was an error in the previous figure legend that has been now corrected. The unboxed sites in green are only found in the HEK293 expressed protein. We have modified the figure and its legend to make it easier for the reader and clarify the meaning of each labelling.

The Excel file provided does not have a legend supplied.

We apologize for the oversight and have included legends for the table in the new version. They can be found after the Supp. Fig. legends in the manuscript

1.- for the HEK293

Phosphorylation sites identified by LC-MS/MS in TLK2 expressed in HEK293 cells. Sheet 1 “Phospho(STY)Sites” contains the raw-output of the MaxQuant search output. Sheet 2 “P(STY) Sites TLK2” is a subset of sheet 1 with only sites in TLK2. Sheet 3 “TLK2 Norm P-Int” shows the log-transformed and normalized (row-wise median subtraction) intensities of all phosphorylation sites. Sheet 4 “TLK2 Median Occupancy” contains occupancy values for all phosphorylation sites represented with median and standard deviation per construct.

2.- for the E.coli

Phosphorylation sites identified by LC-MS/MS in TLK2 expressed in *E. coli*. Sheet 1 “Phospho(STY)Sites” contains the raw-output of the MaxQuant search output. Sheet 2 “P(STY) Sites TLK2” is a subset of sheet 1 with only sites in TLK2. Sheet 3 “TLK2 Filtered-41Sites NormInt” contains the log-transformed and normalized (row-wise median subtraction) intensities of all phosphorylation sites with missing values imputed as specified in the supplementary methods. Sheet 4 “NormInt Distribution” confirms the normal distribution of the intensity values. Sheet 5 “KDom Filter-11Sites NormInt M41” shows only phosphorylation sites in the kinase-domain.

Occupancy values are given but how are these determined.

The method used to determine the values is referenced to the MaxQuant original paper (Ref.70, Cox, J. & Mann, M. *MaxQuant enables high peptide identification rates, individualized p.p.b.-range mass accuracies and proteome-wide protein quantification. Nat Biotechnol* **26**, 1367-72, 2008) in materials and methods in the section “*Processing and analysis of mass spectrometry raw data*”

To further clarify this point we have included two new references (70-71 for MaxQuant and 72 for the occupancies in the new version)

Ref.71 Tyanova, Stefka, Tikira Temu, and Juergen Cox. 2016. “The MaxQuant Computational Platform for Mass Spectrometry-Based Shotgun Proteomics.” *Nature Protocols* 11 (12): 2301–19.

Ref.72 Olsen, Jesper V., Michiel Vermeulen, Anna Santamaria, Chanchal Kumar, Martin L. Miller, Lars J. Jensen, Florian Gnad, et al. 2010. “Quantitative Phosphoproteomics Reveals Widespread Full Phosphorylation Site Occupancy during Mitosis.” *Science Signaling* 3 (104): ra3.

Frequently the pattern of tryptic cleavage of a dephosphorylated site is different from the cleavage around a phosphorylation site. Although SD values are listed the significance has not been calculated. While an average occupancy of 47%, the average SD is 12%.

The high variability is inherent to occupancy values, which is why we chose to perform our analyses based on phosphopeptide intensities, instead. This is discussed in more detail in the following reference:

Dephore, Noah, Kathleen L. Gould, Steven P. Gygi, and Douglas R. Kellogg. 2013. "Mapping and Analysis of Phosphorylation Sites: A Quick Guide for Cell Biologists." Molecular Biology of the Cell 24 (5): 535–42.

The point of requesting a TOF profile of the TLK2 was to see if the dominant species had on average a few phosphates or 10 or more. This would be a trivial experiment for the authors to undertake.

We have performed the ESI-TOF analysis requested by the referee. In this experiment (see Fig. below). There are no distinct dominant species and we can observe that on average we have 20-23 phosphorylations per TLK2.

Figure: ESI-TOF of deltaN-TLK2 protein. The average number of phosphorylations is normally distributed without showing any distinct dominant phosphorylation states.

One of the concerns about the numerous phosphorylation sites reported here is the possibility that many are artefacts of over expression in E coli or HEK293 cells.

The referee is still concerned about the expression method, we would like to comment that:

1.- It is important to mention that the levels of expression in HEK293 cells are many orders of magnitude lower than in E.coli. In addition, to avoid higher levels of expression, cells were collected one day after transfection. No overexpression can be detected at this early collection time, normally cells are collected 3-4 days after transfection for overexpression experiments.

2.- The papers that reported the sites in the Phosphosite web server are from high-throughput datasets. In addition, none of them analysed purified TLK2. They report phosphorylations found in cellular proteomes or tissue proteomes. Therefore, we find reasonable that the

analysis of the purified protein achieves deeper phosphosite coverage because the sample is very much enriched.

3.- We show that many of the phosphosites can be also observed after expression of TLK2 in HEK293, thus supporting that they can be generated in the native environment of human cells with much lower rates of expression than in E.coli.

If one inspects the p-sites recorded in Phosphosite the most prominent (number of reports from HTP studies) occur in the first 1-185 residues with few in the region 191-771. There are only 8 sites (S223, Y259, S752, S753, S763, S769, S770 and S771) recorded in Phosphosite compared with ~40 sites reported here. The occupancy of the shared sites between Phosphosite and the authors are S223 (67%), S770 (81%) and S771 (45%). In order to gain insight into the significance of the phosphorylation sites identified it would be important to know whether they occur under native/in vivo conditions.

Regarding this point, we have found most of the sites reported for the 1-185 section. About the 191-771 region, it is worth mentioning again that:

1.- The Phosphosite web server reports phosphorylations found in cellular or tissue proteomes. None of the papers used to compile the phosphorylations on TLK2 analysed the purified protein, as we have done in our manuscript. Therefore, it seems logical that in the analysis of the purified protein we find more phosphosites.

2.- The expression of TLK2 in HEK293 cells and the MS analysis of the purified protein has shown that a substantial number of the phosphosites found in E.coli can be also found in HEK293 cells. Therefore, we have shown that they can occur when expressed in human cells, where the protein is expressed in much lower amounts than in E.coli (see point 1 in the previous question).

While our data between bacteria and human cells is largely consistent, we appreciate that this does not indicate that every site occurs under physiological conditions or necessarily has an important functional role. We provide the information of the phosphosites detected and we have mutated 4 of them to test their effect on the activity (see below). This has been shown in the previous and in the updated versions of the manuscript. We hope that in the future we can address the individual or collective functions of most of these phosphosites and their physiological relevance, as is mentioned in Pag. 22 (*Lines523-527*) of our manuscript, but this will take a great deal of time and the generation of many new reagents. Here, we have analysed by mass spectrometry the phosphorylations in TLK2 in a controlled (E.coli) and native (HEK293) systems, and demonstrated that many of the sites that we have identified in the kinase domain are functionally relevant. In addition, these data are complemented in our manuscript with the structural, oligomerisation, biochemical and inhibitor data, plus the analysis of the ID mutations. All these findings are novel.

Second it would be important to know if at least one or preferably a few p-sites had functional roles in vivo.

The reviewer missed that we have mutated 4 of the new phosphosites found in the kinase with clear effects on its activity. These kinase mutants were isolated from HEK293 cells, see Figure 5B and Supp. Fig. 8. The figs. show the effect of mutating these new phosphosites (S569, S617, S686, S695), and an additional site (S659), which was suggested by the crystal structure, on the kinase activity.

Therefore, we have investigated the functional roles of 4 of the new sites found in TLK2 expressed in HEK293 cells. Further characterisation of additional sites will be performed in future work, as is stated in Pag. 21 of this version of the manuscript and also in the previous one.

Without this information the authors are not providing a sufficient advance on the importance of dimerisation and the mechanism of activation.

The reviewer misses that our biochemical assays show that deltaN-TLK2 is a dimer, while the kinase domain is a monomer (Fig. 1b), which displays residual activity for autophosphorylation (Fig 2b) and for phosphorylation of ASF-1 (Fig. 2e), its physiological substrate. To the best of our knowledge this is the first time that is shown that the oligomeric state plays an important role in TLK2 activity and its activation.

Reference 60 *regulation and innovative drug development. Semin Cancer Biol. 2018 Feb;48:1-17. doi: 10.1016/j.semcancer.2017.05.011.*

Thanks for pointing this out, the reference has been updated..